# Equivariant Finite Normalizing Flows

## Abstract

Generative modelling seeks to uncover the underlying factors that give rise to observed data that can often be modeled as the natural symmetries that manifest themselves through invariances and equivariances to certain transformation laws. However, current approaches to representing these symmetries are couched in the formalism of continuous normalizing flows that require the construction of equivariant vector fields—inhibiting their simple application to conventional higher dimensional generative modelling domains like natural images. In this paper, we focus on building equivariant normalizing flows using a finite number of layers. We first theoretically prove the existence of an equivariant map for compact groups whose actions are on compact spaces. We further introduce three new equivariant flows: $G$-Residual Flows, $G$-Coupling Flows, and $G$-Inverse Autoregressive Flows that elevate classical Residual, Coupling, and Inverse Autoregressive Flows with equivariant maps to a prescribed group $G$. Our construction of $G$-Residual Flows are also universal, in the sense that we prove an $G$-equivariant diffeomorphism can be exactly mapped by a $G$-residual flow. Finally, we complement our theoretical insights with demonstrative experiments—for the first time—on image datasets like CIFAR-10 and show $G$-Equivariant Finite Normalizing flows lead to increased data efficiency, faster convergence, and improved likelihood estimates.

## 1 Introduction

Many data generating processes are known to have underlying symmetries that govern the resulting data. Indeed, understanding symmetries within data can lead to powerful insights such as conservation laws in physics that are a direct result of the celebrated Noether's theorem and the understanding of natural equivariances of structured objects such as sequences, sets, and graphs. Representing such geometric priors as inductive biases in deep learning architectures has been a core design principle in many equivariant neural network models leading to large gains in parameter efficiency and convergence speeds in supervised learning tasks (Cohen & Welling, 2016; Weiler & Cesa, 2019). Perhaps more importantly, equivariant models also enjoy a data-independent form of generalization as they are guaranteed to respect the encoded equivariances regardless of the amount of data available during training.

For generative modelling tasks like exact density estimation, the development of equivariant architectures currently requires the careful design of invariant potential functions whose gradients give rise to equivariant time-dependent vector fields. These methods belong to a larger family of models called Continuous Normalizing Flows (CNFs) (Chen et al., 2018) and, while elegant in theory, impose several theoretical and practical limitations. For instance, each equivariant map within a CNF must be globally Lipschitz, and in practice, we need to use off-the-shelf ODE solvers for forward integration which may require hundreds of vector field evaluations to reach a suitable level of numerical accuracy. CNFs are also susceptible to other sources of numerical errors such as computing gradients via backward integration which is prone to producing noisy gradients that lead to computationally expensive training times and inferior final results (Gholami et al., 2019). Moreover, to calculate the log density requires computing the divergence of the vector field which for computational tractability is estimated via the Hutchinson's trace estimator whose variance scales linearly with dimension restricting its applicability to smaller systems in some applications (Köhler et al., 2020). Consequently, such numerical challenges inhibit the simple application of equivariant continuous flows to traditional generative modelling domains such as invariant density estimation over images that are of significantly higher dimensions than previously considered datasets.

Normalizing flows composed of finite layers —i.e. each layer is an invertible map, are an attractive alternative to CNFs as they do not, in general, suffer the aforementioned numerical challenges nor do they require external ODE solvers. However, imbuing symmetries within previously proposed flow layers is highly non-trivial, as for certain architectures the equivariance and invertibility constraints may be incompatible. Furthermore, even if both constraints are satisfied the resulting function class modelled by the flow may not be universal—diminishing its ease of application when compared to non-equivariant finite flows. As a result, one may ask whether it is even possible to build equivariant finite layers that are also invertible such that classical finite normalizing flows (e.g. RealNVP, Inverse Autoregressive Flows, Residual Flows, etc ...) can be imbued with powerful inductive biases that preserve the natural symmetries found in data?

**Present work**. In this paper, we consider the general problem of building equivariant diffeomorphisms as finite layers in an Equivariant Finite Normalizing Flow (EFNF). To do so, we first ask the fundamental question—an open problem despite the existence of continuous equivariant flows—of whether such a map even exists between two invariant measures. We affirmatively answer this open question when both the group $G$ and the space on which it acts are compact. Leveraging our theoretical insights we design $G$-residual and $G$-Affine Coupling Flows that are not only invertible but are also equivariant by construction. Moreover, our $G$-Affine Coupling allows constructing equivariant instantiations of two popular flow architectures in RealNVP (Dinh et al., 2017) and the Inverse Autoregressive Flow (Kingma et al., 2016).

In sharp contrast with past efforts on equivariant flows our proposed layers are designed in the language of Steerable CNNs which defines a very general notion of equivariant maps via convolutions on homogeneous spaces, of which $\mathbb{R}^n$ is only one example. Consequently, our proposed flows are more naturally equipped to handle conventional datasets in generative modelling such as images which are invariant to subgroups of the Euclidean group in two dimensions, $E(2)$. On a theoretical front we study the representational capabilities of $G$-Residual flows, and show that any equivariant diffeomorphism on $\mathbb{R}^n$ can be exactly mapped by a corresponding $G$-residual flow when operating on an augmented space —i.e. zero padding the input. Our key contributions are summarized as follows:

- We propose two novel $G$-equivariant layers: $G$-Affine coupling and $G$-residual using which we build $G$-equivariant finite normalizing flows.

- We prove the existence of an equivariant diffeomorphism between two invariant measures when the group and the space it acts on are compact.

- We study the representational capabilities of our proposed equivariant finite flows. We show, by example, that while $G$-coupling flows cannot be applied to all input data types; in contrast $G$-residual flows are universal. Specifically, we prove the universality of $G$-residual flows by demonstrating that any $G$-equivariant diffeomorphism on $\mathbb{R}^n$ can be exactly represented by a $G$-residual flow.

- We conduct experiments on both synthetic invariant densities and higher dimensional image datasets such as Rotation MNIST and augmented CIFAR10 and highlight the benefits of EFNFs.

## 2 Background and Preliminaries

### 2.1 Equivariance and Linear Steerability

In this section, we briefly review the necessary facts regarding equivariance. First, recall the main definition:

**Definition 2.1.** Let $X$ and $Y$ be two sets with an action of a group $G$. A map $\phi : X \to Y$ is called $G$-equivariant, if it respects the action, i.e., $g \cdot \phi(x) = \phi(g \cdot x), \forall g \in G$ and $x \in X$. A map $\chi : X \to Y$ is called $G$-invariant, if $\chi(x) = \chi(g \cdot x), \forall g \in G$ and $x \in X$.

Modelling equivariant maps becomes important when we have a group action on the input and we want to "preserve" this action during the transformation. For example, if we have an image, we would like its intermediate representations to transform accordingly every time we rotate the image itself. Cohen & Welling (2016) introduced steerable CNNs satisfying this property. In particular, an (ideal) image can be thought as a function $f : \mathbb{R}^2 \to \mathbb{R}^K$, where $K$ is a number of channels. The set of all possible images $\mathcal{F}$ form a vector space.

Having a two dimensional representation of a group $G$, one can define its action on the set of all images via the *induced representation* $[\text{Ind}_G^{(\mathbb{R}^2,+) \rtimes G} R]$ where $G = \text{E}(2)$. Note, that the group $\text{E}(2)$ contain elements $(tg)$ where $t \in (\mathbb{R}^2, +)$ and $g \in \text{O}(2)$. We can then simply write the action on $\mathcal{F}$ by: $[g \cdot f](x) = R_g f(g^{-1}(x - t))$, where $R_g \in \text{GL}(\mathbb{R}^c)$ is a representation and $c$ is the number of channels. Clearly, input images transform as scalar fields in which case $\rho(g)$ is the trivial representation, but intermediate layers in an equivariant network may transform under other types such as the regular representation. If one considers a feature space $\mathcal{F}'$ (a vector space with $G$-action), then a convolution layer $\phi : \mathcal{F} \to \mathcal{F}'$ is called steerable if it is $G$-equivariant; it's kernel $\kappa$ can be built using irreducible representations of $G$ (Weiler & Cesa, 2019).

## 2.2   Normalizing Flows

In this section, we recall the most relevant constructions of Normalizing Flows and set notations. A more extensive overview of Normalizing Flows is given in (Kobyzev et al., 2020; Papamakarios et al., 2019). To model a target density one can choose a simple base density and then transform it with a parameterized diffeomorphism (a differentiable invertible map whose inverse is also differentiable). A *normalizing flow* is such a way to model a diffeomorphism such that its Jacobian determinant and inverse are easily computable (i.e., faster than $O(n^3)$). More formally, starting from a sample from a base distribution, $z \sim q(z)$, and a diffeomorphism $f : \mathbb{R}^n \to \mathbb{R}^n$, then the density $p(z')$ of $z' = f(z)$ is determined by the chain rule as:

$$\log p(z') = \log q(z) - \log \det \left| \frac{\partial f}{\partial z} \right|. \tag{1}$$

There are multiple ways to construct normalizing flows. In this work we consider flows which are compositions of finite number of elementary diffeomorphisms ($f = f_K \circ f_{K-1} \ldots \circ f_1$)—so-called *discrete flows*—and whose output density is determined by: $\ln p(z_K) = \ln p(z_0) - \sum_{k=1}^{K} \ln \det \left| \frac{\partial f_k}{\partial z_{k-1}} \right|$. We will also focus on the case when each $f_i$ is an affine flow (Dinh et al., 2017; Rezende & Mohamed, 2015) due to their simplicity and expressiveness in capturing complex distributions.

**RealNVP**. Dinh et al. (2017) introduced real-valued non-volume preserving (RealNVP) flows, where affine coupling layers are stacked together. These layers update a part of the input vector using a function that is simple to invert, but which depends on the remainder of the input vector in a complex way. Formally, given an $n$-dimensional input $x$ and $d < n$, the output $y$ of an affine coupling layer $\mathcal{C} : \mathbb{R}^n \to \mathbb{R}^n$ is given by:

$$y_{1:d} = x_{1:d} \tag{2}$$
$$y_{d+1:n} = x_{d+1:n} \odot \exp(s(x_{1:d})) + t(x_{1:d}). \tag{3}$$

One can immediately write its inverse which itself is another coupling layer. For the remainder of the paper we will use $\mathcal{C} = \text{CONCAT}(\mathcal{C}_1, \mathcal{C}_2)$ to refer to coupling layer operations on the first partition (identity map) and second partition of the input (equation 3) respectively.

**Inverse Autogressive Flows**. Affine Coupling flows using the RealNVP architectures enjoy computationally efficient evaluation and sampling but often require a longer chain of transformations in comparison to other flow-based models to achieve comparable performance. An extension of the RealNVP architecture which transforms the entire input in a single pass while iterating over all feature indices is the Inverse Autoregressive Flow (IAF) model (Kingma et al., 2016). Specifically, an IAF $\mathcal{I}$ model with Affine Coupling can be defined over each index $i$, using again scale and translation networks $s$ and $t$, as follows:

$$\mathcal{I}(x)_i = s_i(x_{<i}) \cdot x_i + t_i(x_{<i}). \tag{4}$$

As the $i$-th input depends on all previous indices $< i$ the Jacobian of an IAF layer is lower triangular and can be computed in linear time.

**Residual Normalizing Flows**. An orthogonal approach to designing expressive finite normalizing flows is to instead consider residual networks as a form of Euler discretization of an ODE.

$$x_{t+1} = x_t + h_t(x_t) \tag{5}$$

Here, $x_t$ represents the activations at a given layer $t$ (or time). A sufficient condition for invertibilty then is $\text{Lip}(h_t) < 1 \; \forall t = 1, \ldots, T$. A Resnet whose $h_t(x_t)$ satisfies the invertibility condition is known as an i-ResNet in the literature Behrmann et al. (2019).

In practice, satisfying the Lipschitz constraint means constraining the spectral norm of any weight matrix —i.e. $\text{Lip}(h) < 1$ *if* $||W_i||_2 < 1$. While i-ResNets are guaranteed to be invertible there is no analytic form of the inverse. However, it can be obtained by a fixed-point iteration by using the output at a layer as a starting point for the iteration. Also, the computation of the Jacobian determinant can be estimated efficiently.

## 3 Existence of Equivariant Diffeomorphisms

In this paper we are most concerned with the generative modelling of symmetric densities defined over Euclidean spaces. More specifically, assume that we have a group $G$ acting on $\mathbb{R}^n$ and our goal is to model a $G$-invariant density $p$. Adopting the normalizing flows approach, one needs to understand how to pick the base density $q$ and how it should be pushed to the desired target $p$. The basic result opening the topic of equivariant normalizing flows is the following theorem.

> **Theorem 1.** *(Papamakarios et al., 2019; Köhler et al., 2020, Theorem 1) Let $p$ be the density function of a flow-based model with transformation $\phi : \mathbb{R}^n \to \mathbb{R}^n$ and base density $q$. If $\phi$ is $G$-equivariant map and $q$ is $G$-invariant density with respect to a group $G$, then $p$ is also $G$-invariant density on $\mathbb{R}^n$.*

The proof can be found in (Papamakarios et al., 2019) with an analogous result in (Köhler et al., 2020). Theorem 1 gives us a general recipe to construct an equivariant normalizing flow by choosing an appropriate invertible map $\phi$ that is equivariant with respect to $G$. A small technical consideration is the choice of base distribution $q$ which must be invariant with respect to $G$, for example the standard normal distribution is invariant to rotations about the origin and reflections.

The next question one could ask is whether such a construction of an equivariant map $\phi$ is always possible. In particular, let $p$ and $q$ be two $G$-invariant densities. Does there always exists a $G$-equivariant map $\phi$, that pushes forward one density into another (i.e., $p\mathbf{dx} = \phi_*(q\mathbf{dx})$)? We can prove a more restricted result in the case when the group $G$ is compact and the ambient space is not $\mathbb{R}^n$ but a compact manifold $\mathcal{M}$.

> **Theorem 2.** *Let $G$ be a compact group with a smooth action on a connected compact smooth orientable manifold $\mathcal{M}$ with or without boundary. Let $\mu$ and $\nu$ be two $G$-invariant volume forms representing the given orientation. Assume that $\int_{\mathcal{M}} \mu = \int_{\mathcal{M}} \nu$. Then there exists a $G$-equivariant diffeomorphism $\phi$, such that $\phi^* \mu = \nu$.*

*Proof Sketch.* The full proof is detailed in §A while here we provide a proof sketch for the case when $\mathcal{M}$ is a closed manifold. The proof can be obtained as an equivariant modification of the Moser's trick (Moser, 1965). First, the equality of integrals implies the existence of the form $\eta$, such that $\nu = \mu + d\eta$. Without loss of generality, we can assume that the form $\eta$ is $G$-invariant (if not, we can average it by a group action and consider a new form instead). Then, we can connect the volume forms $\nu$ and $\mu$ by a segment: $\mu_t = \mu + td\eta$ for $t \in [0, 1]$. Note that $\mu_0 = \mu$ and $\mu_1 = \nu$. We want to find a 1-parameter continuous family of diffeomorphisms (an isotopy) $\{\phi_t\}_{t \in [0,1]}$, such that:

$$\phi_t^* \mu_t = \mu_0. \tag{6}$$

As the manifold is compact, an isotopy can be generated by the flow of a time-dependent vector field $v_t$ (Spivak, 1999, Chapter 5). As deducted in the computation given in equations 21 - 24 in the Appendix §A, the desired equation 6 will be satisfied if:

$$i_{v_t} \mu_t = \eta. \tag{7}$$

Because $\mu_t$ is non-degenerate, we can solve this equation pointwise. As a result, we obtain a unique smooth vector field $v_t$. The compactness of $\mathcal{M}$ allows us to integrate $v_t$ in the flow $\phi_t$. The integration of $v_t$ will result in an $G$-equivariant diffeomorphism as required. □

Theorem 2 guarantees the existence of an equivariant diffeomorphism that pushes forward a base density to any desired target. Crucially, this means that if $\phi$ is within the representation capability of a chosen flow model then it justifies our goal of modelling invariant densities via equivariant diffeomorphisms. Note that an alternative proof of this fact is given in Katsman et al. (2021). They proved the existence of the diffeomorphism by integrating a constructed vector field in the direction of decreasing KL divergence between the source and the target distributions which requires fixing a Riemannian structure over the manifold. In contrast, our result and proof follows a different logic and is more geometric.

In section §4.2 we show that the equivariant diffeomorphism $\phi$ can always be represented exactly using a $G$-residual flow operating on an augmented space.

## 4 Constructing $G$-Equivariant Normalizing Flows

The existence of a $G$-equivariant diffeomorphism—while providing a solid theoretical foundation—gives no instruction on how we can easily construct such maps in practice. We now outline and demonstrate practical methods for building $G$-equivariant maps in the language of discrete normalizing flows. Specifically, we are interested in crafting normalizing flow models that take an invariant prior density to an invariant target density where the invariance properties are known *a priori*. More precisely, this means that each invertible function, $f_i : \mathbb{R}^n \to \mathbb{R}^n$, in our flow must additionally be an equivariant map with respect to a prescribed group $G$ acting on $\mathbb{R}^n$. Mathematically, this requires each $f_i$ to satisfy the following transformation law:

$$f_i(R_g x) = T_g f_i(x), \quad \forall x \in \mathbb{R}^n, \forall g \in G,$$

where $R_g$ and $T_g$ are two representations of the group element $g \in G$.

For the remainder of the paper we will take the group $G$ to be a subgroup of the Euclidean group in $n$-dimensions $E(n) \cong (\mathbb{R}^n, +) \rtimes O(n)$ which can be constructed from the translation group and the orthogonal group in $n$-dimensions via the semi-direct product. For instance, natural images—despite being objects in $\mathbb{R}^n$—may transform along isometries of the plane $\mathbb{R}^2$ (e.g. rotations, reflections, and translations) which are captured by the group $E(2)$. Consequently, to understand how data transforms under prescribed equivariance constraints it is equally important to understand how data can be best represented. In §4.1 we outline one possible avenue for representing data as a combination of a chosen base manifold such as $\mathbb{R}^2$ and natural fibers (e.g. channels in an image) that assign a value to every point on the base. We then exploit this construction in §4.4 and §4.2 to build $G$-Affine coupling and $G$-residual flows respectively while proving the universality of the latter in §4.3. Finally, we close the section with a construction of linear equivariant maps which are more of theoretical than practical interest.

### 4.1 Exploiting Geometric Structure

In classical approaches to normalizing flows including prior work on equivariant CNFs it is customary to treat each input, as a point residing in some $n$-dimensional space by vectorizing the input $\hat{x} = \text{Vec}(x)$. However, a seemingly innocuous operation like vectorizing, without corresponding constraints on the model, also destroys exploitable geometric structure within the data—e.g. rotation equivariance—hampering the overall generative modelling task. Fig. 1 illustrates this phenomena on an image of a planar circle which is rotated by $\pi/2$ rotations. Clearly, a planar-circle is invariant to any continuous rotations in $\text{SO}(2)$, but when discretized into four quadrants transforms according to the finite subgroup $C_4$. We can now treat this $2d$-planar circle as a point in $\mathbb{R}^4$ by discretizing and labelling

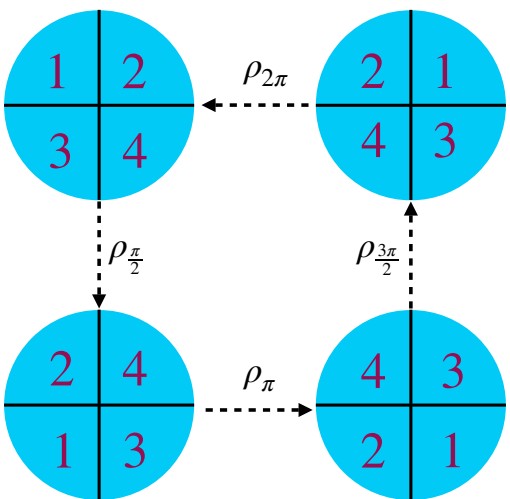

Figure 1: All possible actions of the group $C_4$ on a 2D image of a circle. Each quadrant is labelled from $1-4$.

each quadrant from 1 to 4. Now any rotation in $\mathbb{R}^4$ can be modelled as an element $g \in \mathrm{SO}(4)$, but notice that the action of $\pi/2$ planar rotations corresponds to specific permutation labels for the quadrants—i.e. $R_g \in \mathrm{GL}(4)$ is a permutation matrix. In fact, these are the *only* permissible permutations out of a possible of 4! permutations. This important observation dictates that any equivariant normalizing flow should assign equal likelihoods of the 2$d$-planar circle to all transformations under $C_4$ but not all possible permutations of the quadrants.

How can we build normalizing flows that are capable of exploiting the rich underlying geometry found in data modalities like images? We leverage the theory of steerable CNNs (Cohen & Welling, 2016) which provides a thorough treatment of building equivariant linear maps on homogenous spaces. More prominently, it allows us to reason over different types of geometric structures present in data in the language of associated vector bundles. For example, in the case of equivariant generative modelling of images, we can consider $\mathbb{R}^2$ to be the base space and each channel (e.g. RGB) as a scalar field that transforms independently. In fact, the theory of steerable G-CNN's is applicable not just to scalar fields but also to vector and tensor fields making them an ideal tool to study general equivariant maps. The principal benefit of this design choice is that any equivariance constraint on a given layer can be represented as a convolution kernel with an analogous constraint on a given linear operator. We now introduce two novel equivariant layers that serve as building blocks to constructing $G$-equivariant normalizing flows.

### 4.2 $G$-**Residual Normalizing Flows**

As demonstrated we can convert regular coupling transformations to be equivariant by modifying a vanilla coupling layer to a $G$-coupling layer. This raises the question of whether such a layer exists for the other families of normalizing flows, namely Residual Flows. We now show that this is indeed possible by introducing a $G$-Residual layer. Let us consider residual networks of the form:

$$\phi^i(x) = x + h^i(x)$$

where each sub-network $h^i : \mathbb{R}^d \to \mathbb{R}^d$ at layer $i$ is parametrized by it's weight matrix $W^i$. A composition of maps $\phi^i$'s —i.e. $\phi = (\phi^1 \circ \phi^2 \circ \cdots \circ \phi^T)$ is then a deep residual network. If the map $h$ also satisfies the condition $\mathrm{Lip}(h) < 1$ then each $\phi$ is also invertible Behrmann et al. (2019). If $h$ is additionally equivariant with respect to a group $G$ then the map $\phi^i$ —and by extension $\phi$— is also an equivariant map and we call the equivariant map $\phi^i$ a $G$-Residual layer.

> **Proposition 1.** Let $\phi^i : \mathbb{R}^n \to \mathbb{R}^n$ be a $G$-residual layer as defined above. Let $R$ be a representation for the group $G$. If $h^i : \mathbb{R}^n \to \mathbb{R}^n$ is a $G$-equivariant map then, $\phi^i$ is also a $G$-equivariant map —i.e. $\phi^i(R_g x) = R_g \phi^i(x)$ with respect to the group $G$.

*Proof.* We begin by writing the result of first transforming the input to a $G$-residual layer as follows:

$$\phi^i(R_g x) = R_g x + h^i(R_g x) \tag{8}$$
$$= R_g x + R_g h^i(x) \tag{9}$$
$$= R_g(x + h^i(x)) \tag{10}$$
$$= R_g \phi^i(x) \tag{11}$$

$\square$

It is important to note that our chosen way of enforcing invertibility by controlling the Lipschitz constant of every layer is fully compatible with equivariance. That is point-wise multiplication on the weight matrices with $c/\sigma$ commutes with any group representation $R_g$.

### 4.3 Representation Capabilities of $G$-Residual Flows

For any $G$-equivariant flow it is natural to ask about its representational power in the space of functions and distributions. Unfortunately, it is well known that i-resnets, neural ODEs, and as a result, conventional residual flows are unable to approximate any diffeomorphism without the use of auxiliary dimensions —i.e. zero-padding (Zhang et al., 2020; Dupont et al., 2019). For example, the function $f(x) = -x$ cannot be approximated by an i-resnet. An analogous question for $G$-residual flows is whether they can approximate any $G$-equivariant diffeomorphism on $\mathbb{R}^n$. We now show it is possible to construct—in a similar manner to Zhang et al. (2020)—a $G$-Residual Flow to exactly build any $G$-equivariant diffeomorphism. To do so, we first extend the action of $G$ to the augmented space in a trivial manner.

Given an action of a group $G$ on a vector space $V$, then for any other vector space $V'$, we can define a $G$-action on $V \oplus V'$ by: $g \cdot [v, v'] \mapsto [g \cdot v, v']$. This helps us to extend a $G$-action to the padded space, where $V = V' = \mathbb{R}^n$. We can then extend Theorem 6 of Zhang et al. (2020) to the equivariant case.

> **Theorem 3.** *Assume that $G$ acts on $\mathbb{R}^n$. For any $G$-equivariant Lipschitz continuous diffeomorphism $\phi : \mathbb{R}^n \to \mathbb{R}^n$ there exists a $G$-residual flow on the padded space $\psi : \mathbb{R}^{2n} \to \mathbb{R}^{2n}$, where the $G$ action is extended to the padding of $\mathbb{R}^n$ as above, such that $\psi([x, 0]) = [\phi(x), 0]$ for any $x \in \mathbb{R}^n$.*

*Proof Sketch.* The full proof is outlined in §B. As Zhang et al. (2020) show there exists a Residual flow on the padded space $\psi : \mathbb{R}^{2n} \to \mathbb{R}^{2n}$, such that $\psi([x, 0]) = [\phi(x), 0]$. We must now show that this flow is equivariant with respect to the extended $G$-action on $\mathbb{R}^{2n}$. Indeed by $G$-equivariance of $\phi$ and the definition of the extended $G$-action we have:

$$\psi(g \cdot [x, 0]) = \psi([g \cdot x, 0]) = [\phi(g \cdot x), 0]$$
$$= [g \cdot \phi(x), 0] = g \cdot [\phi(x), 0]$$
$$= g \cdot \psi([x, 0]).$$

$\square$

Theorem 3 explicates that a $G$-residual operating on an augmented space is sufficiently powerful to exactly map a $G$-invariant prior to a $G$-invariant target. What Theorem 3 does not say, however, is the ease of optimization and the resulting training dynamics due to the interplay of the Lipschitz and equivariance constraints which are needed for invertibility and guaranteeing the pushforward to be an invariant density.

### 4.4 $G$-Coupling Normalizing Flows

Coupling layers form the backbone of countless normalizing flow architectures due to their simplicity in computing the change in volume as well as having an analytic inverse. Thus it is tempting to consider whether such coupling layers can be made $G$-equivariant. To construct such a flow we need to impose some restrictions on the flow function and the representation of the group. We outline these in the following proposition below.

> **Proposition 2.** Let $\mathcal{C} : \mathbb{R}^n \to \mathbb{R}^n$ be a coupling layer with functions $s$ and $t$ as defined as follows.
>
> $$y_{1:d} = x_{1:d} \tag{12}$$
> $$y_{d+1:n} = x_{d+1:n} \odot s(x_{1:d}) + t(x_{1:d}). \tag{13}$$
>
> Also let $R$ be a $n$-dimensional representation of the group $G$. Assume that $n = 2d$ and $R$ is a completely reducible diagonal permutation representation: $R(g) = (R_g, R_g)$, where $R_g$ is a permutation $d \times d$ matrix. If $s$ and $t$ are $G$-equivariant then the coupling layer $\mathcal{C}$ is $G$-equivariant.

*Proof.* The equivariance condition for the flow $R(g)\mathcal{C}(x) = \mathcal{C}(R(g)x)$, $\forall g \in G$) written explicitly gives:

$$R(g)\mathcal{C}(x) = \begin{pmatrix} R_g x_{1:d} \\ R_g x_{d+1:n} \odot s(R_g x_{1:d}) + t(R_g x_{1:d}) \end{pmatrix} \tag{14}$$

Equation 14 outlines which restrictions are sufficient on the functions $t$, $s : \mathbb{R}^d \to \mathbb{R}^d$ to obtain an equivariant coupling flow. Using the assumption on the representation $R$, one has:

$$R_g \left( \mathcal{C}_2(x) \right) = R_g x_{d+1:n} \odot s(R_g x_{1:d}) + t(R_g x_{1:d}).$$

Where $\mathcal{C}_2$ is a coupling layer operating on the second partition of the input vector as described in equation 3. Since permutation matrices satisfy the following identity $R_g(x \odot y) = (R_g x) \odot (R_g y)$, it is sufficient to take $s$ and $t$ to be $G$-equivariant. Using this and the fact that the identity is trivially equivariant the overall coupling layer $C$ is also equivariant. $\qquad\square$

**Remark 1.** From equation 14 we already see that the representation $R$ cannot be irreducible. To preserve equivariance we need a non-linearity equivariant to permutations (for example, element-wise exp as in equation 3). Furthermore, because of the non-commutativity of the Hadamard product with general matrix multiplication, $R$ cannot be any representation. Indeed, this is the key rationale used by Köhler et al. (2020) to justify the negative claim that $G$-equivariant coupling flows do not exist. However, as we prove if we employ the permutation representation then the Hadamard product is commutative and as result, we have a $G$-equivariant coupling layer. The immediate consequence of needing to use permutation representations is that $G$ must be finite which eliminates compact groups such as SO(2). However, in practice due to discretization and aliasing of signals, finite subgroups of E(2) under the regular representation—which are permutations themselves—are a large class of groups that can be modelled using $G$-coupling flows.

Our proposed definition of the $G$-coupling layer can be seen as the most general equivariant coupling layer and is in fact a strict generalization of previous efforts when $G$ is taken as the permutation group (Rasul et al., 2019; Biloš & Günnemann, 2021). The $G$-coupling layer, while being equivariant, is limited by the fact the representation of the group $R$ cannot be irreducible. In practice, we can take representations of the group independently for each channel attached to the base space (e.g. RGB channels in an image where the base space is $\mathbb{R}^2$). However, when such a decoupling is not possible a $G$-coupling equivariant flow cannot learn the desired target density.

### 4.5 $G$-**Inverse Autoregressive Flows**

In a similar manner to $G$-Coupling flows we can construct Inverse Autoregressive Flows that respect the desired symmetry constraint. However, unlike non-equivariant IAF models to bake in non-trivial symmetries, we consider $k$ equal partitions of the input which allows us to define a layer in a $G$-equivariant IAF flow.

> **Proposition 3.** Let $\mathcal{I}(x)_i : \mathbb{R}^d \to \mathbb{R}^d$ be the $i$-th block transformation of an IAF layer with scale and translation functions $s$ and $t$ as defined above. Also let $R$ be a $n$-dimensional representation of the group $G$. Assume that $n = k \cdot d$ and $R$ is completely reducible diagonal permutation representation: $R(g) = (R_{g_1}, R_{g_2}, \cdots, R_{g_k})$, such $R_{g_i}, i \in [k]$ is a permutation $d \times d$ matrix. If $s$ and $t$ are $G$-equivariant then the IAF layer $\mathcal{I}$ is $G$-equivariant.

*Proof Sketch.* The $G$-equivariance of an Affine Coupling layer applied $k$-times for each partition $i \in [k]$ of the input to the IAF layer. $\qquad\square$

### 4.6 **Invertible Equivariant Linear Maps**

In the previous sections, we considered imbuing equivariance into the coupling and residual flows yet it is well known in the literature that a linear map that is equivariant must be a convolution with steerable kernels (Cohen et al., 2018). One may then ask if such maps can also be invertible enabling the construction of linear equivariant flows. Here we choose to present the theory in its abstract form with no insight into

practical instantiations—i.e. the group $G$ is not any specific group like E(2), and the reader may choose to skip this section and move directly to §6 to avoid interruptions in the flow of exposition.

To achieve the goal of building linear equivariant maps let us first consider the space of all linear maps, $\mathrm{Hom}_G(\mathcal{F}_n, \mathcal{F}_{n+1})$ not necessarily equivariant, between an arbitrary layers $n$ and $n+1$. The set of equivariant linear maps, also called intertwiners, forms a vector space and can be constructed under the following constraint on $\mathcal{H} := \mathrm{Hom}_G(\mathcal{F}_n, \mathcal{F}_{n+1})$,

$$\mathcal{H} : \{f \in \mathrm{Hom}_G(\mathcal{F}_n, \mathcal{F}_{n+1}) \mid f R_{n,g} = R_{n+1,g} f \quad \forall g \in G\}.$$

It is well known that a continuous linear map under mild assumptions can be written as a continuous kernel $\kappa : \mathbb{R}^n \times \mathbb{R}^n \to \mathbb{R}^{K_n \times K_{n+1}}$:

$$[\kappa \cdot T](x) = \int_{\mathbb{R}^n} \kappa(x,y) T(y) dy \tag{15}$$

Here $K_m$ and $K_{n+1}$ are the dimensionality of the feature (field) attached at each point in the respective layer's feature space. When $\mathcal{F}_n$ and $\mathcal{F}_{n+1}$ are taken to be induced representations the equivariance constraint results in a one-argument kernel which can be thought of a convolution like integral (Cohen et al., 2018; Weiler et al., 2018; Weiler & Cesa, 2019). Thus the kernel is subject to the following linear constraint:

$$\kappa(gx) = R_{n+1,g} \kappa(x) R_{n,g}^{-1}, \tag{16}$$

where $R_{n,g}$ is a representation of $G$, and whenever numerically feasible the kernel can be built via basis functions using irreducible representations of $G$.

**Invertible Kernels**. An observant reader may also recognize equation 15 as the definition of an integral transform of which the input is a function. Such transformations need not be invertible but whenever an inverse exists the inverse transform must satisfy:

$$\int_{\mathbb{R}^n} \kappa^{-1}(y-x) \kappa(y'-x) dy = \delta(y-y') \tag{17}$$

A few well-known integral transforms include the Fourier and Laplace transform each of which is completely determined by the choice of kernel and interestingly both transforms are invertible. It is worth noting for invertibility to hold it is necessary for the kernel to be non-zero everywhere, for example, such a condition is needed in the standard convolution theorem for the Fourier transform. Thus these kernels not only satisfy the linear constraint in equation 16 but also the invertibility constraint equation 17.

**Matrix Exponential Equivariant Flows**. An alternative to finding a kernel that is simultaneously equivariant and invertible is to impose invertibility on the overall operation. As demonstrated by Hoogeboom et al. (2020) when the convolution operation is expanded as matrix multiplication with an appropriately expanded kernel $z = \kappa * x = K\vec{x}$, acting on a vectorized input $\vec{x}$ invertibility corresponds to the invertibility of $K$. In general expanded equivariant kernels need not be invertible but their matrix exponential of $K$ —i.e. $e^K$ are guaranteed to be invertible and also preserves equivariance Hoogeboom et al. (2020); Xiao & Liu (2020). Thus the matrix exponential-based flow is also an example of an equivariant linear flow as the matrix exponential must be numerically approximated using a truncated power series.

## 5 Related Work

Imbuing normalizing flow models with symmetries was first studied concurrently (Köhler et al., 2020) and (Rezende et al., 2019). The former approach utilizes continuous normalizing flows (CNF) and is closest in spirit to the $G$-residual flows proposed in this paper. The second approach uses the Hamiltonian formalism from physics and decouples a system into its generalized position and momentum for which a finite-time invertible solution can be found using Leapfrog-integration. As a result, such an approach can be seen as the specific but continuous time instantiation of the proposed $G$-coupling flow. Flow-based generative models have also seen applications with regard to specific symmetry groups, most notably the symmetric group which manifests itself through permutation invariance. For example, when data is comprised of sets —e.g.

a dataset of $N$ point clouds— and the permutation invariance is known as exchangeability of finite data. At present, the only flow-based models that handle this specific case utilize a coupling-based transform (Rasul et al., 2019; Bender et al., 2020) but can be thought of as another instantiation of the $G$-coupling layer. Outside of permutations, equivariant flows have also been constructed for $E(n)$ (Satorras et al., 2021) which in contrast to this work requires global equivariance and is more suitable to model application domains such as molecular dynamics where a global coordinate frame can be easily assigned. Lastly, building equivariant generative models beyond flows has also been a nascent but active direction of research with equivariant score matching (De Bortoli et al., 2022) and diffusion models (Hoogeboom et al., 2022; Xu et al., 2022; Igashov et al., 2022) which may applications in molecular generation.

Finally, the study of equivariances in theoretical physics has a long and celebrated history. Recently, the application of flow-based generative models has risen in prominence in the study of such systems like sampling in Lattice Gauge Theory (Kanwar et al., 2020), the $SU(N)$ group (Boyda et al., 2020), and most recently Katsman et al. (2021) construct equivariant manifold flows—extending euclidean space equivariance results in Papamakarios et al. (2019)—for isometry groups of a given manifold.

## 6 Experiments

We evaluate the ability of our Equivariant Finite Flows on both synthetic toy datasets and higher dimensional image datasets in Rotation MNIST and CIFAR-10. For baselines, we use classical RealNVP-based coupling flows and Residual flows which are non-equivariant but are trained with heavy data augmentation sampled uniformly at random from the group. In practice, when constructing $G$-Residual flows the input and output of a layer must be a scalar field which transforms according to the trivial representation which at first might appear limiting. However, the function $h$ inside a residual layer itself can have intermediate representations that are not scalar fields (e.g. we can use regular, irreps, etc. . . ) as long as the output of $h$ also maps to a scalar field. Such a design principle—where intermediate layers can include more complex representations than the trivial one—is common in steerable neural networks (Weiler & Cesa, 2019). Moreover, we also point out that our equivariance theory is constructed for $n$-dimensional spaces and to apply this to natural images we embed our data into $\mathbb{R}^2$ with channels being the fibers. As a result, this means that we are able to use steerable-convolutions for subgroups of $O(2)$ which acts on the base space of $\mathbb{R}^2$ but because of the embedding in $\mathbb{R}^n$ equivariance is preserved.

### 6.1 Synthetic Experiments

We first consider a toy dataset of a mixture of 8-gaussians and 4-concentric rings in $\mathbb{R}^2$. The empirical distribution for each dataset contains both rotation and reflection symmetries and as a result, we would like the pushforward of an $G$-invariant prior (e.g. bivariant normal distribution) to remain invariant under the action of $G$. For fair comparison, we allocate a parameter budget of  2k to each model and train for 20k steps using Adam with default hyperparameters (Kingma & Ba, 2014). Note that, while conventional coupling flows could also be employed here their equivariant counterparts preserve equivariance along the channel dimensions, thus hindering their application on single channel data. In Fig 2 we visualize the density of each for non-equivariant and equivariant residual flows for groups $C_{16}$ and $D_{16}$. We observe that the equivariance constraint, in such low parameter regimes, enables the learned density not only to respect the data symmetries but also to produce sharper transitions between high and low-density regions as found in the target density.

### 6.2 Image Datasets

We now consider density estimation over images as found in Rotation MNIST and CIFAR-10. Note that CIFAR-10 by itself does not have rotation symmetry but reflection symmetry and as a result—for fair comparison—we construct an augmented dataset where each input is transformed via an element from $G$ uniformly at random (Weiler & Cesa, 2019). In fig 3 we visualize generated samples from $G$-Residual Flows trained on three variations of the MNIST dataset equipped with $D_8, C_{16}$, and the translation group $T$ symmetries respectively. In table 1 we report bits per dim values for non-equivariant and $G$-residual

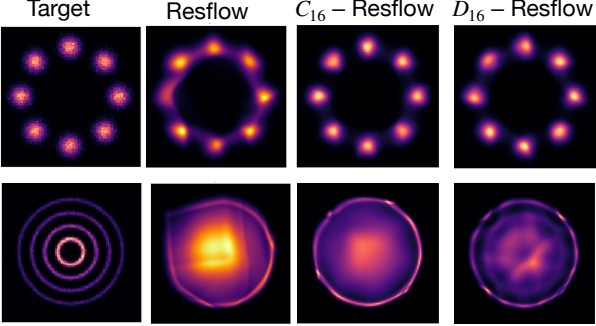

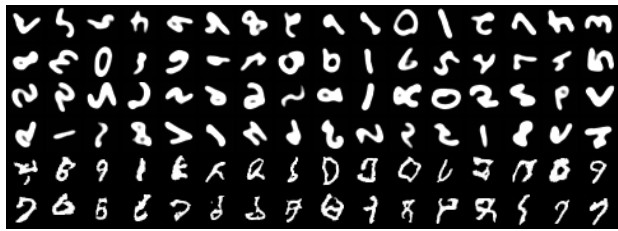

Figure 2: Density Estimation over Toy Data in $\mathbb{R}^2$ using non-equivariant and $G$-residual flows. For 8 Gaussians we notice that $G$-equivariant Resflows have less smearing of the probability mass on individual modes. For the spirals dataset we observe that all Resflows with $2k$ parameters struggle to completely separate the high and low-density regions but $D_{16}$ Resflow is able to respect the data symmetry better.

Figure 3: Generated samples from $G$-Equivariant Residual Flows. The top two rows illustrate samples from the group $D_8$ while the middle two rows the group is $C_{16}$. In the first four rows we notice rotated digits about the center of the image and a few reflections about the vertical axis. The bottom two rows illustrate the translation group $T$ implemented by considering the input space as a Torus and vanilla convolutions with circular padding. Here we notice generated digits splicing at the boundary of the image indicating translation symmetry.

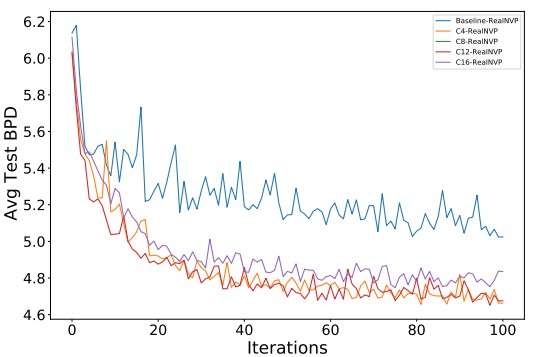

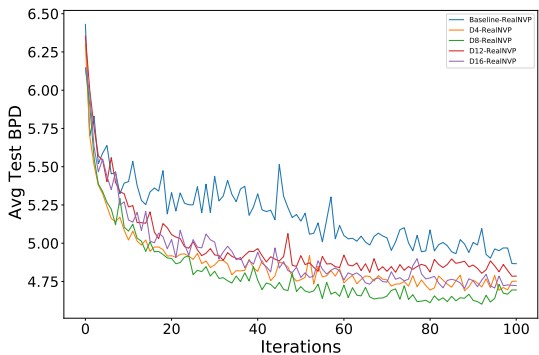

Figure 4: Ablation study on augmented CIFAR-10 using $G$-Coupling flows with discrete subgroup of $SO(2)$ (Left figure) and $O(2)$ (Right figure).

flows as well their respective parameter counts. We observe that $G$-residual flows produce slightly worse bits per dim but compensate with large gains in parameter efficiency —i.e. 227% and 195% respectively on Rotation MNIST and CIFAR-10. We hypothesize that $G$-residual flows while universal are harder to optimize due to the potentially destructive interference between the Lipschitz and equivariance constraints. Moreover, due to the equivariance constraints on the kernel each forward pass requires us to rebuild the kernel which adds a computational overhead inhibiting larger model sizes. Furthremore, in the specific case of $G$-residual flows these kernels must also be Lipschitz (—i.e. The matrix $W$ used in the convolution operation must be Lipschitz) which is enforced via spectral or induced matrix norm. Simultaneously satisfying both the Lipschitz and equivariance constraints leads to an increased training overhead as both are themselves optimization problems internal to the general optimization problem of maximizing the log-likelihood of the data. In practice, we also find that the equivariance constraint and Lipschitz constraint can often be antagonistic with each other as forcibly enforcing equivariance can force the network to increase the Lipschitz constant to compensate which can lead to training instability. In these cases, it is important to more aggressively enforce the Lipschitz constraint after each iteration which is in contrast to vanilla Residual Flows that trade off enforcing this constraint after every $N$ iterations in favor of training speed.

Table 1: Density Estimation over images in Rotation MNIST and CIFAR-10. Each cell reports bits per dim as well as the total number of parameters used.

| Model | Rotation MNIST | CIFAR-10 |
|---|---|---|
| RealNVP | - | $4.87 \mid 6.2 \times 10^6$ |
| $C_4$-RealNVP | - | $4.66 \mid 1.09 \times 10^6$ |
| Resflow | $1.65 \mid 9.78 \times 10^3$ | $3.87 \mid 2.83 \times 10^4$ |
| $G$-Resflow | $2.07 \mid 2.99 \times 10^3$ | $4.04 \mid 9.57 \times 10^3$ |

For CIFAR-10 we additionally perform an ablation study using $G$-coupling flows for subgroups of SO(2) and O(2). Note, that because our $G$-coupling flows must use the permutation representation in theory we can only model discrete subgroups of SO(2). Consequently, as the exact equivariance for $G$-coupling flows to permutation representations is a rigid constraint on model expressivity we implement soft equivariance using regular $G$-convolutions in place of vanilla convolutions. We report the average group test bits per dimension which compute the mean bits per dimension assigned to a test datapoint across all elements in a finite subgroup (e.g. $C_{16}$) of a larger continuous group. Fig. 4 illustrates $G$-coupling flow models on finite subgroups of SO(2) while Fig. 4 right repeats the same procedure for O(2). As observed in both ablation experiments we find that all $G$-coupling flows converge significantly faster and to lower average group bits per dim than the non-equivariant coupling flow. Interestingly, we also observe increasing the equivariance constraint—e.g. $C_4$ vs. $C_{16}$—leads to improved test bits per dim values highlighting the benefits of modelling invariant densities using equivariant maps.

## 7 Conclusion

In this paper, we study the problem of building equivariant diffeomorphisms on Euclidean spaces using finite layers. We first theoretically prove the existence of an equivariant map for compact groups with group actions on compact spaces laying principled foundations for the modelling of invariant densities using equivariant flows. We introduce Equivariant Finite Normalizing Flows which enable the construction of classical normalizing flows but with the added guarantee of equivariance to a group $G$. To build our EFNFs we introduce $G$-coupling and $G$-residual layers which elevate classical coupling and residual flows to their equivariant counterparts. Empirically, we find $G$-residual flows enjoy significant parameter efficiency but also lead to a small drop in performance in density estimation tasks over images. $G$-coupling flows on the other hand are limited in their applicability to domains which contain more than a single channel (e.g. RGB images) but achieve faster convergence to lower average test bits per dim on augmented CIFAR-10. While we proved the existence of equivariant maps between invariant densities on compact spaces many data domains with symmetries are in fact non-compact and proving the existence of an equivariant map in this setting is a natural direction for future work.

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

# A    Existence of the equivariant map

## A.1    Proof of Theorem 2.

*Proof.* Here we provide the necessary details to the sketch of the proof we gave earlier. The proof can be obtained as an equivariant modification of the Moser's trick (Moser, 1965; Banyaga, 1974).

First, we need to recall two facts: 1) for any compact connected closed oriented smooth manifold, its top cohomology group is one-dimensional; 2) for any compact connected oriented smooth manifold with nonempty boundary, its top cohomology group is zero (for the proof of these two statements see, for example, (Weintraub, 2014, Theorem 8.3.10)). Hence, because we are additionally given that $\int_M \mu = \int_M \nu$, there exists $\eta \in \Omega^{(n-1)}(M)$, such that $\nu = \mu + d\eta$.

In case $M$ has a nonempty boundary, we automatically get an additional condition on the form $\eta$ that its integral over the boundary must vanish:

$$0 = \int_M \nu - \int_M \mu = \int_M d\eta = \int_{\partial M} \eta. \tag{18}$$

The last equality is Stoke's theorem.

Without loss of generality, we can assume that the form $\eta$ is $G$-invariant. Indeed, because of the naturality of the action (denote it $R$), one has:

$$dR_g^* \eta = R_g^* d\eta = R_g^*(\nu - \mu) = \nu - \mu, \ \forall g \in G. \tag{19}$$

The last equality holds because the forms $\nu$ and $\mu$ are given to be $G$-invariant. Then if we fix the Haar measure on $G$ (we can do it because $G$ is compact), we can average the form $\eta$, and consider this new form instead of $\eta$. By construction, it is $G$-invariant.

Then, connect the volume forms $\nu$ and $\mu$ by a segment: $\mu_t = \mu + td\eta$ for $t \in [0,1]$. Clearly, $\mu_0 = \mu$ and $\mu_1 = \nu$. By construction, each $\mu_t$ is a $G$-invariant $n$-form and $\int_M \mu_0 = \int_M \mu_t$.

We want to find a 1-parameter continuous family of diffeomorphisms (an isotopy) $\{\phi_t\}_{t \in [0,1]}$, such that

$$\phi_t^* \mu_t = \mu_0. \tag{20}$$

Indeed, if we substitute $t = 1$ it will give the desired result. As the manifold is compact, an isotopy can be generated by the flow of a time-dependent vector field $v_t$ (Spivak, 1999, Chapter 5). Let's differentiate equation 20 with respect to $t$. The right-hand side will give zero, while the left-hand side is:

$$\frac{d}{dt}(\phi_t^* \mu_t) = \phi_t^*(\mathcal{L}_{v_t} \mu_t + \frac{d}{dt}\mu_t), \tag{21}$$

where $\mathcal{L}$ is Lie derivative. Note that this equation is a chain rule. Recall Cartan's formula:

$$\mathcal{L}_v = di_v + i_v d \tag{22}$$

where $i_v$ is an interior product. Using this, the fact that $d\mu_t = 0$ (because it is a top form) and the computation $\frac{d}{dt}\mu_t = d\eta$, we have:

$$\frac{d}{dt}(\phi_t^* \mu_t) = \phi_t^*(di_{v_t} \mu_t + d\eta) = \phi_t^* d(i_{v_t} \mu_t + \eta). \tag{23}$$

This will be equal to zero, if

$$i_{v_t} \mu_t + \eta = 0. \tag{24}$$

Because $\mu_t$ is non-degenerate, we can solve equation 24 pointwise. As a result, we obtain a unique smooth vector field $v_t$. The compactness of $M$ allows us to integrate $v_t$ in the flow $\phi_t$. In case when the manifold $M$ has a boundary, we additionally get that the restriction of the vector field on a boundary is zero: $v_t|_{\partial M}$ Banyaga (1974), hence $\phi_t|_{\partial M} = Id$.

Since $\mu_t$ and $\eta$ are $G$-invariant, so is the vector field $v_t$. The integration of $v_t$ will result in an $G$-equivariant diffeomorphism. □

## B    Representation Capabilities of $G$-Residual Layer

*Proof.* First, we need reproduce the existential result of Zhang et al. (2020). We show that for any Lipschitz continuous diffeomorphism $\phi : \mathbb{R}^n \to \mathbb{R}^n$ there exists a Residual flow on the padded space $\psi : \mathbb{R}^{2n} \to \mathbb{R}^{2n}$, such that $\psi([x, 0]) = [\phi(x), 0]$. Denote the Lipschitz constant of $\phi$ by $L$. Also denote $T = \lfloor L + 1 \rfloor$. Consider the function: $\delta(x) = \frac{\phi(x) - x}{T}$. Clearly $\delta(x)$ is smooth and its Lipschitz constant is less than one. Hence, the map of the padded space $\mathbb{R}^{2n} \to \mathbb{R}^{2n}$ given by:

$$\psi_0 : [x, y] \mapsto [x, y] + [0, \delta(x)] \tag{25}$$

is an i-ResNet. In particular $\psi_0([x, 0] = [x, \delta(x)]$.

Now let us consider the map

$$\psi_i : [x, y] \mapsto [x, y] + [\frac{yT}{T+1}, 0], \text{ where } i = 1, \ldots, T + 1. \tag{26}$$

The residual part has Lipschitz constant less than one, hence this map is also an i-ResNet. By definition of $\delta$, we have:

$$\psi_{T+1} \circ \cdots \psi_1 \circ \psi_0([x, 0]) = [\phi(x), \delta(x)]. \tag{27}$$

Finally, consider the map:

$$\psi_{T+2} : [x, y] \mapsto [x, y] + [0, -\delta(x)]. \tag{28}$$

Similarly to $\psi_0$ it is an i-ResNet.

Overall, denote the composition of all $\psi_i$ where $i = 0, \ldots, T + 2$ by $\psi$. Then we showed that it is a ResFlow (as a composition of i-ResNets). And by construction, $\psi([x, 0]) = [\phi(x), 0]$.

Now we must show that if the diffeomorphism $\phi$ is additionally taken to be $G$-equivariant, then the ResFlow $\psi$ that we constructed above is also $G$-equivariant with respect to the extended $G$-action on $\mathbb{R}^{2n}$. Indeed by $G$-equivariance of $\phi$ and the definition of the extended $G$-action we have:

$$\begin{aligned}
\psi(g \cdot [x, 0]) = \psi([g \cdot x, 0]) &= [\phi(g \cdot x), 0] \\
&= [g \cdot \phi(x), 0] = g \cdot [\phi(x), 0] \\
&= g \cdot \psi([x, 0]).
\end{aligned}$$

$\square$

## C    Experimental Details

### C.1    $G$-Residual Flows Architecture

The architecture we used for our $G$-residual flows is built off the codebase in the original Residual flows architecture (Chen et al., 2019). To maintain equivariance we replace the LipSwish activation function with a point-wise non-linearity (e.g. Pointwise-ReLU (Weiler & Cesa, 2019)). We also remove the input LogitTransform layer as we found it leads to training instability. Finally, we also convert the activation normalization layer to an equivariant one and we use it before and after each residual block. Each residual connection consists of:

$$\text{ReLU} \to 3 \times 3 \text{ Conv} \to \text{ReLU} \to 1 \times 1 \text{ Conv} \times \text{ReLU} \times 3 \times 3 \text{ Conv}$$

Each convolution layer contains 128 hidden units and while the input and output transform under the trivial representation intermediate layers transform under the regular representation. The main differentiating factor from vanilla residual flows is that kernels in each of the convolution layers are built with respect to specific kernel constraints of a given group as described in the appendix of Weiler & Cesa (2019). It is important to note the number of effective parameters are the ones that correspond to the learnable basis

coefficients for each convolution kernel and not all the weights themselves. We now outline the specific architectures for each dataset used in the main paper:

**RotationMNIST**.
$$\text{Image} \rightarrow 8 \times \text{ResBlock} \rightarrow \text{Image}$$

**FlipRotMNIST**.
$$\text{Image} \rightarrow 8 \times \text{ResBlock} \rightarrow \text{Image}$$

**https://www.overleaf.com/projectCifar10**. For Cifar10 as observed in (Weiler & Cesa, 2019) we use $5 \times 5$ convolution kernels as opposed to $3 \times 3$ in the MNIST datasets.

$$\text{Image} \rightarrow 12 \times \text{ResBlock} \rightarrow \text{Image}$$

## C.2 $G$-**Coupling Flows Architecture**

Unlike the non-equivariant RealNVP architecture which makes use of various masking patterns and the multi-scale architecture we cannot directly use these in building $G$-Coupling flows as it would break equivariance. Instead, $G$-Coupling flows employ channel-wise masking such that the group acts on each channel independently. For our non-linearity we use the PointWise ReLU in a similar manner to $G$-Residual Flows. Each $G$-Coupling block consists of:

$$s : 4 \times [3 \times 3 \text{ Conv} \rightarrow \text{InnerBatchNorm} \rightarrow \text{ReLU}] \rightarrow 3 \times 3 \text{ Conv} \rightarrow \text{InnerBatchNorm} \rightarrow \text{ReLU}$$
$$t : 4 \times [3 \times 3 \text{ Conv} \rightarrow \text{InnerBatchNorm} \rightarrow \text{ReLU}] \rightarrow 3 \times 3 \text{ Conv} \rightarrow \text{InnerBatchNorm} \rightarrow \text{ReLU}$$

Each convolution layer contains 128 hidden units and the last convolution layer in every block brings the input type to be a scalar field—i.e. a valid image.

**Cifar10**. For Cifar10 we use the following architecture and train with a weight decay of $1e - 4$

$$\text{Image} \rightarrow 8 \times G - \text{Coupling Layer} \rightarrow \text{Image}$$

