# OpenReview forum: "Equivariant Finite Normalizing Flows"
_TMLR — Rejected by TMLR_

### Review · Reviewer_deMB · 2022-09-26

**Summary Of Contributions:**

This paper proposes a variety of ways to extend so-called "discrete" normalizing flows to respect various symmetries through equivariance. In particular, the paper primarily focuses on generalizing residual, coupling, and autogressive normalizing flow architectures to be equivariant to some group $G$.

**Requested Changes:**

Theoretical Results
----------------------
* A stronger version of Theorem 2 appears in Katsman et al. 2021 (in particular they use heat flow to construct a gradient flow that couples probability measures on compact manifolds). Furthermore, as Theorem 2 uses Moser's coupling, the actual result can be seen by inspection and does not warrant such an in-depth proof. In particular, the Moser vector field build is built from the distributions to be coupled, and, by doing so, one can see that invariant distributions lead to an equivariant vector field. Finally, the authors should note that Theorem 2 is heavily divorced from the actual applications of the paper. In particular, the results are purely geometric and crucially do not generalize to compact sets in R^n (in particular compact sets with nonzero volume in R^n are not compact manifolds but rather compact manifolds with boundary).
* Proposition 1 follows from simple linear algebra facts. It should not be included as a proposition; perhaps a remark.
* Theorem 3 follows directly from definitions as well. Should be shortened similar to Proposition 1.
* The results of Proposition 2 should note the fact that the permutation condition is pretty restrictive. In particular, if the representation of each group element is a permutation matrix, this implies that the group G is discrete (and perhaps there are other conditions I'm missing). This is needed since many previous results such as Kohler et al. 2020 operate on continuous Lie Groups like SO(3).
* Similar to the above statement, Proposition 3 should also have this note.

Unclear
---------
* For the background introduction to RealNVP, i'm assuming that "D" variable should be "n".
* The authors should mention that Theorem 1 relies on G being a subgroup of E(n), a fact that they later use.
* I'm unsure what the purpose of Section 4.1 is. I can see that it serves as some motivating factor for equivariance, but this should either be included in a non-contribution section or made more concise.
* For proposition 2, I'm assuming that the n=2d should be replaced with D=2d as in the background section.
* The authors should precisely introduce the groups C16, D16, etc... in the experimental section.

Experiments
--------------
* The results in Figure 2 are not convincing. For example, the results for the C16 and D16 results for concentric circles do not appear to be invariant with respect to the groups. This leads me to suspect at least several implementation bugs.
* The authors should also mention more about which architectures they use for the equivariant networks. In particular, this is the nonidentity part in the residual flows and the s, t networks in the realnvp like architecture. In particular, they should be specific about which transformations are used and the hidden dimension of each.
* The results in Figure 3 are not very convincing, as the generated images are not very good. Since this is MNIST, one should expect better image generation.
* The results in Table 1 are not convincing. In particular, since one wants to have a lower BPD, I'm unsure how ResFlow being better than G-ResFlow on higher parameter count shows a contribution.

**Strengths And Weaknesses:**

Strengths
-----------
+ The paper provides a relatively thorough treatment of equivariant flows and how to construct equivariant versions of classical architectures.
+ The paper provides good theoretical justification of said models with some proofs of universality/representation power.

Weaknesses
--------------
- I have several problems with the theoretical results (elaborated below).
- The paper, as written, is often confusing / under-specificied in several areas.
- The experiments do not provide a good empirical backing of the central claims.

---

> ### Author Response · Authors · 2022-10-12
> **Rebuttal Part 1/2**
>
> We thank the reviewer for the comments and feedback. We especially appreciate that the reviewer felt that our paper provided a thorough treatment of equivariant flows and how to construct equivariant architectures. We also thank the reviewer for acknowledging that our paper provides good theoretical justification for our proposed models along with results on universality and representational power of $G$-residual flows. We now address the key clarifications points below.
>
>
> **Q. Relation of Theorem 2 and Katsman et. al 2021**
>
> We appreciate the reviewer's concern for our Theorem, especially given a similar result appears in Katsman et. al 2021. We would like to disagree with the characterization of the reviewer that the result in Katsman et. al 2021 is stronger as first we note that their setting requires a closed (compact) Riemannian manifold while in our setting we require only a closed compact manifold—not necessarily Riemannian. Furthermore, our proof follows a different line of reasoning which we argue is simpler to follow as noted by the Reviewer. We argue that the simpler formulation of the proof is a strength of the result rather than a counter-point.
>
> The reviewer makes a good observation that Theorem 2’s result is not directly applicable to $\mathbb{R}^n$ which we use for experiments. Despite this difference in settings, we find that building equivariant finite flows does provide tangible practical benefits such as improved parameter efficiency. While the setting of our existence theorem is for compact manifolds we found the paper by Greene and Shiohama (1979) “Diffeomorphisms and volume-preserving embeddings for noncompact manifolds” where Moser’s theorem was extended to noncompact manifolds with boundaries. However, it is highly non-trivial to use the same technique to generalize to the equivariant case. We believe that elevating our existence theorem to non-compact manifolds is an extremely important topic deserving of its own paper in a separate math venue as it requires non-trivial arguments from differential geometry.
>
> **Q. Proposition 1 follows from simple facts**
>
> We agree with the reviewer that the proof is indeed easy to come up with but we include this proof for the convenience of the reader, to fix notation, and finally for the completeness of the result all of which we believe is important within the context of a paper.
>
> **Q. Proof of Theorem 3.**
>
> The proof for Theorem 3 is also included for completeness. We also note that the importance of theorem 3 is decoupled from the simplicity of its proof. In particular, with our universality result for $G$-residual flows we can hope to always learn an equivariant map between two invariant densities using a residual flow architecture in appropriately padded space. Such a fact is not obvious at all and the proof only appears simple conditioned on the important result of Zhang et. al 2020.
>
> **Q. Permutation representation is restrictive**
>
> This restriction is only for $G$-coupling flows and not for $G$-residual flows. We explain this point transparently in Remark 1 of the paper and in fact this highlights the difficulty of building $G$-coupling flows that are expressive enough.
>
> **Q. Proposition 3 should also have this note**
> Please see the previous remark for our thoughts on permutation representations. We agree with the reviewer that this point should be further clarified for $G$-IAFs and we will do so in the updated draft of the manuscript.
>
> **Q. The authors should precisely introduce the groups C16, D16**
>
> We thank the reviewer for their suggestion, we will update the manuscript to include a description of these subgroups.
>
>
> **Q.  Results in Table 1 are not convincing.**
>
> We acknowledge the reviewer's concern. We argue that the results in table 1 show that we can achieve similar results using much fewer parameters. It is important to note that these image datasets are not exactly equivariant but instead with an only subset of the group (e.g. we cannot ask for a digit 6 to be invariant with respect to the full O(2) group otherwise we cannot distinguish between a digit 9). In practice, this constraint hurts likelihood estimation as we really need not the full equivariance constraint and our experiments serve to demonstrate the ability to build equivariant flows rather than achieve SOTA.
>
> **Q.  The authors should also mention more about which architectures they use for the equivariant networks.**
>
> We will include an appendix that outlines all of our architectures for all models. We also plan to release code after the review process.

---

> > ### Author Response · Authors · 2022-10-12
> > **Rebuttal Part 2**
> >
> > **Q. The results for the C16 and D16 results for concentric circles do not appear to be invariant with respect to the groups.**
> >
> > This can be attributed to the discretization error which result in approximate equivariance in steerable CNNs which leads to a density that bleeds out. In particular, the discretization errors arise due to building equivariance kernels beyond the $C_4$ group as it is impossible to achieve exact equivariance for $O(2)$ on $\mathbb{R}^2$. This phenomenon is well known artefact in the literature and is outlined in detail of Appendix Appendix E.3 in Ferbach et. al 2022 [1. In practice “approximate” equivariance is already a powerful inductive bias which is what we seek to demonstrate through our experiments.
> >
> >
> > **Q. Since this is MNIST, one should expect better image generation.**
> >
> > The reviewer makes an astute observation regarding the bits/dim results we report. For equivariant tasks we wish to highlight that the standard dataset is not MNIST but instead MNIST12k as introduced in Weiler and Cesa [2]. This dataset contains on12k samples and is smaller than MNIST and as a result the results are not directly comparable to the bits/dim numbers. Note that the RotMNIST and FlipRotMNIST datasets are derived from MNIST12k but with the group transformations randomly applied to the inputs. Due to the smaller size of the dataset we expect a small dip in generation performance as well.
> >
> >
> > [1] Ferbach, Damien, Christos Tsirigotis, and Gauthier Gidel. "A General Framework For Proving The Equivariant Strong Lottery Ticket Hypothesis." arXiv preprint arXiv:2206.04270 (2022)
> >
> > [2] Weiler, Maurice, and Gabriele Cesa. "General e (2)-equivariant steerable cnns." Advances in Neural Information Processing Systems 32 (2019)

---

> > > ### Comment · Reviewer_deMB · 2022-10-24
> > > **Further Questions**
> > >
> > > Re: concentric circles
> > >
> > > Sure, but this is somewhat of a large limitation, as equivariant CNFs tend to show much higher fidelity (up to numerical solver error).
> > >
> > > Re: MNIST image generation
> > >
> > > I'm not so much concerned about the bits/dim more so as the images that are generated.

---

> > ### Comment · Reviewer_deMB · 2022-10-24
> > **Further Questions**
> >
> > ### Re: Theorem 2
> >
> > I believe both theorems require the manifold to be Riemannian. In particular, it is used to define volume forms (for the central statement of pushforwards of measures used in both theorems) and the gradients/divergence operators (that both theorems use to construct the final flow).
> >
> > Concerning theorem 2 applicability: since the theorem is introduced to "guarantees the existence of an equivariant diffeomorphism that pushes forward a base density to any desired target", it should be applicable to the setting of R^n; since it's not, this does't support the central claim. Note that the linked paper is technically more general than need be (we are only working on compact sets in R^n for universal approximation as otherwise neural networks aren't powerful enough, not general noncompact Riemannian manifolds), so such generality is unneeded.
> >
> > ### Re: Permutation Representation
> >
> > If both IAFs and coupling layers have this restriction, I think this represents a severe limitation of the thesis of the paper (building equivariant discrete normalizing flow layers). Furthermore, this is also made more conspicuous since the strongest results are built for residual networks (which can be thought of as discretization of CNFs).
> >
> > ### Re: Table1
> >
> > But the results are not similar for resflow, defeating the purpose of showing fewer parameters (e.g. maybe to get similar results you need to use a similar number of parameters). Furthermore, if this experiment only "serve[s] to demonstrate" the ability to build equivariant flows, which experiment shows that equivariant flows are useful for real-world high dimensional tasks on which CNFs struggle more with?

---

> > > ### Author Response · Authors · 2022-10-26
> > > **Re: Theorem 2 and Permutation Representation concerns**
> > >
> > > We thank the reviewer for their comments on our rebuttal regarding theorem 2. We agree that the theorem from Katsman et al. 2021 requires the Riemannian structure, otherwise, gradients/divergence operators could not be defined. However, our proof does not need such operators. To define a volume form one does not need a Riemannian structure either. In fact, as the reviewer might agree all one needs is the orientability of the manifold (which we assumed). Specifically, we can rely on the following well-known theorem:
> > >
> > > **Theorem:**
> > > An n-dimensional smooth manifold $\mathcal{M}$ is orientable if and only if $\mathcal{M}$ admits a nowhere vanishing smooth $n$-form. (for a proof please see: http://staff.ustc.edu.cn/~wangzuoq/Courses/18F-Manifolds/Notes/Lec23.pdf)
> > >
> > > As such we would like to politely push back against the assertion that the two theorems (ours and Katsman et. al 2021) are the same.
> > >
> > > With regards to the applicability of our theorem to $\mathbb{R}^n$, we agree with the reviewer that a compact manifold, of course, is different than $\mathbb{R}^n$. While we believe that it is maybe possible to extend our theorem to the case of manifolds with boundary
> > > by potentially modifying the general Moser trick from Green and Shiohama (1979) [1] paper we argue that our theoretical result is still relevant and important. Specifically, it is not uncommon in deep learning for theory to have a slight departure from the experimental setting(e.g. the non-convergence guarantees of SGD to global optimums) and if anything this highlights the difficulty of obtaining theoretical results in this space. Consequently, we believe our theorem offers new results that are not in the literature (compact manifolds vs. Riemannian manifolds) and has a more geometric proof which is simpler and is always valuable from a purely mathematics point of view.
> > >
> > > **IAF and Coupling flows use Permutation Representations**
> > >
> > > As we have outlined in our rebuttal the choice of permutation representation highlights the innate difficulty in building $G$-coupling flows. This restriction pinpoints the reason why initially researchers favored the continuous normalizing flow framework to build equivariant flows as deriving sufficient constraints on the architecture is non-trivial. Furthermore, it is important to note that the regular representation is a permutation representation which is precisely what is used to build $G$-Conv networks, and they have been shown to work better in terms of raw performance and from a stability point of view compared to kernels for other representation types---e.g. irreps, quotient reps---Weiler and Cesa 2021 [2]. As a result, this is not as severe a limitation as one might expect as we can model finite subgroups of $O(2)$.
> > >
> > > [1] Greene, R. E., & Shiohama, K. (1979). Diffeomorphisms and volume-preserving embeddings of noncompact manifolds. Transactions of the American Mathematical Society, 255, 403-414.
> > >
> > > [2] Weiler, M., & Cesa, G. (2019). General e (2)-equivariant steerable cnns. Advances in Neural Information Processing Systems, 32.

---

> > > > ### Comment · Reviewer_deMB · 2022-10-26
> > > > **Re: Re: Theorem 2 and Permutation Representaiton**
> > > >
> > > > # Theorem 2
> > > >
> > > > ## Riemannian metric vs volume form
> > > >
> > > > Note that one can always recover a Riemannian structure that agrees with a volume form. In particular, if the volume form is $v dx_1 \wedge \dots \wedge dx_n$ then the Riemannian metric can be taken to $\sqrt[n]{v}  I$ where g is a smooth function on the manifold (the argument is suppressed for notational purposes) and $I$ is the metric which acts by $I(dx_i, dx_j) = \delta_{ij}$ (identity with respect to the vectors). This works since $v > 0$ as it is part of a volume form.
> > > >
> > > > From the above, both theorems are operating under the same conditions. In particular, given a manifold with only a volume form, one could just instantiate a Riemannian structure and then derive the same result (that the group of volume preserving diffeomorphisms is nonempty).
> > > >
> > > > ## Applicability
> > > >
> > > > Concerning theory vs experimental settings: most deep learning theory papers give theoretical results that only hold given some pre-specified set of pre-conditions. While it is often rare to expect these conditions to work in practice, they sometimes can, at which point the theorem can be applied. Theorem 2 is crucially different since there is no condition which makes it applicable to the experimental setting (flows on $R^n$).
> > > >
> > > > Concerning value of theorem: I don't think the statement is stronger than what exists in the literature (as mentioned above). I also don't think it's simpler. Something simpler would be taking Moser's coupling (used in Moser Flow) and showing that it produces an equivariant vector field (which it does by simple equivariance rules). The proof is probably more geometric, but I don't think this is necessary valuable (especially since the core method already exists in Lec 24 of the linked lecture notes).
> > > >
> > > > # Permutation Representation
> > > >
> > > > I'm not sure how the choice of permutation representation proves that "deriving sufficient constraints on the architecture is non-trivial". Currently, it just shows that the proposed method is not able to handle the general cases. Again, since the paper wants to show that discrete flow architectures can be made group-invariant, a negative result like this should be noted. Furthermore, it is a rather severe limitation, as finite groups are the simplest groups to work with in representation theory (when compared to other general lie groups).

---

> > > > > ### Author Response · Authors · 2022-10-27
> > > > > **Re:Re:Re Theorem 2 and Permutation Representation**
> > > > >
> > > > > **Applicability of Theorem 2**
> > > > >
> > > > > We are grateful to the reviewer for raising the issue of the applicability of Theorem 2 for manifolds with boundary. We have now updated the Theorem statement in our latest edit and our proof now handles this case and we are able prove this theorem in more generality. Consequently, we can omit the assumption that the manifold $\mathcal{M}$ is closed and can deal with compact manifolds with boundary. The modification is minor and we realized that all constructions we had before work (thanks to the paper by A. Banyaga (1974) [1]). We updated the manuscript. We believe that the reviewer's applicability concern is now no longer a problem as just like the Katsman et. al 2021 paper we can handle compact subsets in $\mathbb{R}^n$.
> > > > >
> > > > >
> > > > > **Riemannian metric vs volume form**
> > > > >
> > > > > That is definitely true that any smooth manifold can be endowed with a Riemannian form (just using partition of unity and locally euclidean property). However, if a fact (in our case, existing of a diffeomorphism) could be proved without that, the proof is usually considered more general. Similar to linear algebra---one prefers to prove theorems without fixing a basis in the most invariant form.
> > > > >
> > > > > **“The proof is probably more geometric, but I don't think this is necessary or valuable (especially since the core method already exists in Lec 24 of the linked lecture notes).”**
> > > > >
> > > > > We strongly disagree with this statement. Lecture 24 loc.cit. teaches the standard Moser’s trick. However extending it to the equivariant case is non-trivial and at present doesn’t exist in the literature---as far as we are aware. Hence, we believe our theorem is valuable and enriches the communities understanding of this important result.
> > > > >
> > > > > We would also like to thank the reviewer for pointing out the Katsman et. al 2021 paper to us. We have now updated our manuscript with a citation along with a small discussion to highlight the different approaches taken in the proofs. Furthermore, we believe that the fact in Theorem 2 is very interesting and deserves multiple proofs. In fact, within the mathematics community, it is valuable to have different proofs for the same result if the proof brings new insights to the problem or demonstrates a new way of thinking throw different tools. As a result, we believe both our proof and Katsman et. al 2021's theorem each have their place within the broader literature.
> > > > >
> > > > > **Permutation Representation**
> > > > >
> > > > > As we have agreed with the reviewer already, discrete subgroups are less general than the full continuous groups one might care for. In fact, in our updates to the paper, we modified remark 1 as well as mentioned in several places the limitations of $G$-coupling flows in comparison to $G$-Resflows. We believe we have done enough to explain the importance and difficulty of constructing $G$-coupling flows and having flows that operate on subgroups is not a limitation of the result but rather an important insight in our characterization of various equivariant finite normalizing flows.
> > > > >
> > > > > [1] https://www.e-periodica.ch/cntmng?pid=ens-001:1974:20::57

---

### Review · Reviewer_ZQjr · 2022-09-29

**Summary Of Contributions:**

In this work the authors introduce discrete normalising flows acting on $R^d$ which are $G$-equivariant, leading to $G$-invariant probability distribution---when combined with a $G$-invariant base distribution.
In particular, they extend residual flows to $G$-equivariant residual flows with $G \subseteq E(n)$, and similarly for coupling and autoregressive flows to their $G$-equivariant counterparts with G acting via a permutation matrix, hence finite subgroups $G \nsubseteq O(n)\ltimes T(n)$. They introduce practical implementations for these flows by using on steerable CNNs for the flows components (residual connection or $s$ and $t$ neural networks for coupling and autoregressive flows).
They theoretically show that (on a padded space) G-residual flows are universal approximators.
They also show the existence of G-equivariant flow between two G-invariant densities (when both G and the space it is acting on are compact).
They empirically show the parameter efficiency of the introduced flows in contrast with standard flows, on several image datasats with $E(3)$ symmetries.


**Broader Impact Concerns:**

I do not have concerns on that matter.

**Requested Changes:**

In what follows, I wrote a mixture of questions and suggestions, not necessarily strict 'Requested Changes', but more like opening a discussion on a few (hopefully relevant) points.

- Title: Why 'finite'? I would suggest swapping with 'discrete'
- Section 1
    - "equivariant models also enjoy a data-independent form of generalization as they are guaranteed to respect the encoded equivariances regardless of the amount of data available during training." -> reference?
    - The authors are motivating discrete flows in opposition to continuous flows due to the Lipschitz contraint on the vector field, but isn't this similat to the constraints for ResFlows?
- Section 2
    - The notion of group 'representation' is mentioned without being introduced. 'Note that the defined action is linear' -> I assume that is because *linear* representations are considered. May be useful to clarify this point.
- Section 3
    - Theorem 2
        - As stressed by the authors, the result from Theorem 2 is for compact space, as opposed to the earlier motivation with Euclidean space. Would it be possible to prove a negative result for R^d or would the authors conjecture that it still holds? Where does the current proof breaks?
        - I feel that the current sketch of proof could be enhanced
            - Worth defining what is an 'isotopy'
            - Should state that $\mu_1=\nu$.
            - 'As the manifold is compact, an isotopy can be generated by the flow of a time-dependent vector field' -> reference?
            - I am not certain to fully follow the proof. The existence of a vector which induces a flow is shown, which itself allow to interpolate between the two measures of interest. Then it is stated 'Since $\mu_t$ tand $\nu$ are G-invariant, so is the vector field v t., but how come $\mu_t$ is invariant? Also $\phi_t$ is G-equivaraint iif $v_t$ is G-equivaraint, not G-*invariant*?
- Section 4
    - 4.1: 'any equivariance constraint on a given layer can be represented as a convolution kernel with an analogous constraint' -> on a given *linear* operator
    - 4.2: Does the residual connection forces the output representation of $h_i$ to be of the same type as $x$, that is of type-1? Can this be limiting?
    - 4.4: The constraints on the group representation is that it is a permutation matrix, and since all finite groups are subgroups of the permutation group, this include $C_n$ and $D_n$ groups. Yet, is this group is acting on half of the space, or on the full space? I am struggling to understand how the latter could be true. Perhaps an illustration on a 2d image would be useful to clarify this.
    - 4.6: So Eq 17 is necessary but not sufficient to impose invertability of the linear map? How is the invertibility of the operators $s$ and $t$ in G-coupling flows enforced then?
- Section 5
    - A few relevant work are missing from the related work section. I would suggest mentioning/discussing [A, Appendix L.3] which proposed G-invariant score-based models.
- Section 6
    - 6.1
        - A 'parameter budget of 2k' sounds quite small. In particular, none of the models accurately fit the spirals dataset. Additionaly, although the empirical results show that in this parameter regime the standard ResFlow fails to accurately fit the  mixture of Gaussian dataset, and as such that G-invariant NFs are more parameter efficient, I would be interested in seeing whether the ResFlow would be able to learn the invariance with more parameters.
        - Why not have an O(2)-ResFlow? Then we could check that the density is exactly O(2) invariant.
    - 6.2
        - Similarly here, although the parameter efficiency gain is important, what is preventing from fitting larger G-resflow? Is it a computational bottleneck? Or are numerical instabilities arising (from the constrained choice of non-linearity)?
        - Looking at Table 1 from [B], the reported bits/dim on (non augmented) MNIST and CIFAR-10 are 0.97 and 3.28 respectively, why such a gap with report 3.87 for Resflow in Table 1 of this work?
        - Why not including the G-coupling results from the ablation study in Table 1? Similarly, I would be interested in having results for G-ResFlow for finite subgroups of $O(2)$ and $SO(2)$.
        - Here too the reported bits/dim of 3.49 for Real-NVP from [B] is different from waht we can read in Figure 4 in this work.
        - It would be worth in Table 1, additionally showing results on augmented test dataset, but trained on a non augmented dataset, as a sanity check since then the G-equivariant model should easily outperform its counterpart.
        - What is the reason for not empircally assessing Real-NVP and the introduced G-coupling flows methods?
        - Although this requires more work, it would be of interest to compare the proposed apparoach to G-invariant continuous normalising flows and G-invariant score based models.

[A] De Bortoli et al., Riemannian Score-Based Generative Modeling, 2022.

[B] Ricky T. Q. et al.,Residual flows for invertible generative modeling, 2019.


**Strengths And Weaknesses:**


### Strenghts
First, I found this work pretty well written and quite easy to read.
The paper is well organised, and most concepts are well presented.
It is nice to start with Theorem 2 which shows the existence of G-equivariant flows.
Although reasonnably simple, Proposition 2 which shows a universal approximator result for G-residual flows is welcome.
The proposed flows are both simple and practical, building on prior work for the actual implementation of the flows individual components.

### Weaknesses

Not really a weakness per se, but with regard to writing, I feel that the sketch of proof provided in Section 3 could be enhanced, as some concepts/notations aren't introduced. I have a few questions on this proof in the following section of this review.

I believe that the main weakness of this work is really the empirical demonstration of the practical use of the proposed methods. Section 6 convinced me that the introduced flows can be more parameter efficient than their counterpart but not much more.
I a making more detailed suggestions in the following section, but in short:
  - There seem be some issue with the proposed models when increasing the number of parameters as stated by the authors themselves: 'G-residual flows while universal are harder to optimize'. It is unclear what is the specific issue and whether it is only related to ResFlows or also coupling/autoregressive flows (no Lipschitz constraint)
  - It is lacking some sanity checks such as showing that an O(2)-equivariant flow indeed lead to an O(2)-invariant density, would be particularly relevant in Section 6.1.
  - Similarly, in Section 6.2, would be worth training models on non G-augmented datasets but evaluating them on G-augmented dataset to check that the G-invariant models are over-performing.
  - Lastly, there are limited baselines being assessed, namely ResFlows and Real-NVP, CNFs would be particularly relevant since they are being (rightfully) criticised in the introduction. Training CNFs is well-known to be slow so a fixed computational budget could be set for a fair assessement.
  - Yet, there seems to be a gap between the reported performance (for baselines) and what has been published in the litterature, likely due to the use of smaller networks but that is unspecified.

---

> ### Author Response · Authors · 2022-10-12
> **Rebuttal Part 1**
>
> We thank the reviewer for their detailed comments and helpful feedback on the draft of our paper. We are heartened to hear that the reviewer felt the presentation of the paper to be well organized and that most concepts were well presented. We also appreciate that the reviewer felt that proposition 2 being a universal approximator to be a welcome addition to the paper. Finally, we appreciate that the reviewer felt that our proposed flows were both simple and practical.
> We now respond to the main questions asked by the reviewer below.
>
> **Q.$G$-Equivariant Flows are Harder to Optimize**
>
> We appreciate the reviewers comment in that our phrasing of “harder to optimize” might not have been sufficiently clear which we now clarify. The primary difficulty with regard to optimization lies with $G$-residual flows as increasing the number of parameters means we must rebuild equivariant kernels for every forward pass. This is needed for every $G$-steerable network that makes use of irreps (see Weiler and Cesa 2021 for a detailed discussion on how kernels are built). In the particular case of $G$-residual flows these kernels must also be Lipschitz (---i.e. The $W$ matrix used in convolution must be Lipschitz) which is enforced via spectral or induced matrix norm. Simultaneously satisfying both constraints leads to an increased training overhead as both are themselves optimization problems internal to the general optimization problem of maximizing the log-likelihood of the data. The equivariance constraint and Lipschitz constraint can often be antagonistic with each other as forcibly enforcing equivariance can force the network to increase the Lipschitz constant to compensate which can lead to training instability. In these cases, it is important to more aggressively enforce the Lipschitz constant after each iteration which is in contrast to vanilla Residual Flows who trade off enforcing this constraint after every $N$ iterations in favor of training speed. As a result, we find that for computational tractability it is easier to train smaller $G$-residual networks than Residual Networks found in Chen et. al 2019. We note that $G$-coupling flows do not share this problem as there is no Lipschitz constraint on $W$. We will expand the discussion in the updated draft.
>
> **Q. Sanity Checks using $O(2)$**
> We thank the reviewer for their suggestion. We empirically perform this sanity check for the translation group (see. fig 3). As all residual flows effectively have the same architecture (we will add exact architecture details in a new appendix) where only the group changes we believe this sanity already passes up to the numerical error. We would like to also point out that due to discretization errors that arise due to building equivariance kernels beyond the $C_4$ group it is practically impossible to achieve numerically exact equivariance for $O(2)$ on $\mathbb{R}^2$. This is a well-known phenomenon in the literature (see Appendix E.3 in Ferbach et. al 2022 [1] for a thorough discussion on this matter) and in practice “approximate” equivariance is already a powerful inductive bias which is what we seek to demonstrate through our experiments.
>
> **Q. Training non-$G$ augmented flows but evaluating them on $G$-augmented datasets**
>
> This is an interesting experiment. We did think about this setting but felt this comparison would be a bit of a contrived “straw-man” comparison and were concerned that it might detract from the thrust of the paper.  If the reviewer feels strongly about this point we are happy to include this result. Please do let us know.
>
> **Q. Training Equivariant CNFs over images**
>
> The reviewer makes an interesting observation that our baselines do not include equivariant CNFs. While in principle this could be an interesting baseline there exists a significant barrier to actually materializing this baseline. First, current equivariant CNFs have not been demonstrated to work on symmetries found in images but only on densities on manifolds and dynamical systems. It is non-trivial to build an equivariant vector field on $\mathbb{R}^n$ using steerable convolutions—for example, it is difficult to even construct the representation of the group in $\mathbb{R}^n$ in a practical way. Note that this is needed as the vector field in $\mathbb{R}^n$ has to be equivariant. We bypass this difficulty in our work by using steerable convolutions in $\mathbb{R}^2$ and embed this in a higher dimensional space. A similar strategy might be possible for equivariant CNFs but that would amount to an orthogonal contribution and not a straightforward extension of current equivariant CNFs.

---

> > ### Author Response · Authors · 2022-10-12
> > **Rebuttal Part 2**
> >
> > **Q. Performance gap between baselines in this paper and the literature**
> >
> > The reviewer makes an astute observation that the baseline bits/dim results we report in this paper for our baselines differ from the literature. There are two main reasons for this. In the case of RotMNIST and FlipRotMNIST, the dataset is a subset of size 12k rather than the full MNIST. This dataset was first introduced in Weiler and Cesa 2021 [2] and as a result, we expect worse bits/dim as we are training on a smaller dataset. The second source of discrepancy is due to the fact that we use smaller networks and strip away all performance-enhancing tricks used in these architectures that lead to SOTA numbers. This was done to achieve the fairest comparison to $G$-equivariant networks as we are interested in highlighting the benefit equivariance brings rather than achieving SOTA results. For instance, for coupling flows as in RealNVP we cannot employ the multi-scale architecture as this would break equivariance—i.e. we can only apply masking channel-wise and to the whole image not smaller portions). We understand the reviewer's concern but we reiterate that the goal of our experiments is chiefly to demonstrate the possibility of building $G$-equivariant finite flows while showing their perceived benefits such as improved parameter efficiency and not to push the state of the art results.
> >
> > **Q. Finite vs. Discrete**
> > We thank the reviewer for their suggestion. We think Discrete flows are also equally valid but we feared that this would imply that the data domain was discrete. Given, that there is already literature on discrete flows we opted against this potentially confusing name choice.
> >
> > **Q. “Equivariant models also enjoy a data-independent form of generalization”**
> >
> > Equivariant models have inherent inductive biases that allow them to generalize to transformations that respect group structure in the test set without ever seeing them in the training set. The generalization capabilities of equivariant networks are an exciting theoretical literature and essentially we are able to achieve tighter generalization bounds due to this inductive bias. Please see Sannai et. al 2019 [3] for a more detailed discussion.
> >
> > **Q. The authors are motivating discrete flows in opposition to continuous flows due to the Lipschitz constraint on the vector field, but isn't this similar to the constraints for ResFlows?**
> >
> > The reviewer is correct in pointing out that a similar Lipschitz constraint exists for Residual flow as for CNFs. However, this was not the main thrust in motivating finite flows.  As we write in our introduction CNFs require the usage of ODE solvers for forward integration which may require hundreds of vector field evaluations to reach a suitable level of numerical accuracy and are also susceptible to other sources of numerical errors. For example in practice, this could result in noisy gradients and expensive training times Gholami et. al 2019 [4]. Also, we highlight that the divergence calculation needed for the estimation of log-density via the Hutchinson trace estimator has a variance that scales linearly with the dimension Köhler et al., 2020 [5].
> >
> > **Q. The notion of group 'representation'  is not introduced.**
> > We thank the reviewer for their suggestion. We will update the main text to include this. Note that a group representation is a matrix in $GL(n)$ and as such there is no “non”-linear representation, which does not preclude a non-linear group action.
> >
> > **Q. Theorem 2. On non-compact manifolds and updating the readability of the proof sketch.**
> >
> > We greatly appreciate the reviewer's suggestions on improving the clarity of our proof sketch. We will update the proof sketch in accordance with the reviewer's suggestion.
> > Regarding extending Theorem 2 result to non-compact manifolds we do not currently know of any counter-example that prohibits the extension to this case. Moreover,  we believe that the result still holds and the proof could be potentially generalized. Case in point, we found the paper by Greene and Shiohama (1979) “Diffeomorphisms and volume-preserving embeddings for noncompact manifolds” where Moser’s theorem was extended to noncompact manifolds with boundaries. However, it is highly non-trivial to use the same technique to generalize to the equivariant case. We firmly that proving the generalization is still a worthwhile endeavor but such a proof would be better suited as a separate paper in a math venue.

---

> > > ### Author Response · Authors · 2022-10-12
> > > **Rebuttal Part 3**
> > >
> > > **Q.  Does the residual connection force the output representation to be the same as the input?**
> > >
> > > This is a good observation and it indeed does for the output and input representation to be the same. However, note that Theorem 3 proves the universality of our $G$-residual flows and from a theory perspective this is not an issue. In practice, note that a residual layer utilizes an $h$ network whose internal representation can be arbitrarily complex and can employ different representation types that differ from the input (e.g. regular) despite the output of $h$ requiring the same representation type as the input.
> > >
> > > **Q.  The group representation is a permutation matrix, how does it act on images?**
> > >
> > > For RGB images the group acts independently on each channel while for
> > > gray-scale images, we simply duplicate the channels so that our construction becomes applicable, because now we have two independent channels and can extend the group representation there through the diagonal action.
> > >
> > > **Q. 4.6: So Eq 17 is necessary but not sufficient to impose the invertibility of the linear map? How is the invertibility of the operators of s and t in $G$-coupling flows.**
> > >
> > > We acknowledge the reviewer's concern but we believe that there may be a small source of misunderstanding. For $G$-coupling flows the architecture is itself invertible by construction and are non-linear maps. The discussion in 4.6 centers around equivariant linear maps and this requires the kernel to be invertible (e.g. Fourier transform).
> > >
> > > **Q. A few relevant works are missing from the related work section.**
> > >
> > > Thanks for the reference, it is indeed another way to learn an equivariant map. We will add this reference.
> > >
> > > **Q. A 'parameter budget of 2k' sounds quite small.**
> > > We acknowledge the reviewer's comment and we agree that a much larger ResFlow can fit this dataset but this is not the primary point of our experiments. We wish to highlight the improved parameter efficiency on this synthetic dataset which we derive by utilizing equivariance which is more prominent in smaller-sized networks.
> > >
> > > **Q. What is preventing from fitting larger G-resflow?**
> > > As we mention in our answer above to why $G$-residual flows are harder to optimize there is a computational bottleneck that arises for building larger $G$-resflows as we need to re-build equivariant kernels for every forward pass which leads to significant overhead in training. Post-training this is not a problem because we can always convert a $G$-equivariant network to a standard network by fixing the kernel coefficients. Given the limited computational budget and the main thrust paper being our developed theory we chose to perform demonstrative experiments rather ones than aiming to achieve SOTA.
> > >
> > > **Q. Why not including the G-coupling results from the ablation study in Table 1?**
> > > Thank you for the suggestion, we will update table 1 accordingly.
> > >
> > > **Q. What is the reason for not empircally assessing Real-NVP and the introduced G-coupling flows methods?**
> > > We are not sure we  understand the concern here. We empirically evaluate $G$-coupling flows on CIFAR-10 see Fig 4. Note that the baseline in this figure is RealNVP.
> > >
> > > **Q. Comparing $G$-equivariant finite flows and $G$-invariant score models**
> > > Thanks for the interesting suggestion. An immediate comparison is difficult as it is non-trivial to build a group representation of E(2) in the higher dimensional embedded space. This would require us to build an equivariant vector field in R^n for our chosen groups---at present there is no demonstration of this in the literature and we believe this is a hard problem. We bypass this problem by applying Steerable convolutions that work on R^2 as the space and then embedding this in the higher dimensional space. This may be possible for score and diffusion based models but it is not immediately clear how to do so and we believe this to be an interesting question for future work.
> > >
> > >
> > > [1] Ferbach, Damien, Christos Tsirigotis, and Gauthier Gidel. "A General Framework For Proving The Equivariant Strong Lottery Ticket Hypothesis." arXiv preprint arXiv:2206.04270 (2022)
> > >
> > > [2] Weiler, Maurice, and Gabriele Cesa. "General e (2)-equivariant steerable cnns." Advances in Neural Information Processing Systems 32 (2019)
> > >
> > > [3] Sannai, Akiyoshi, Masaaki Imaizumi, and Makoto Kawano. "Improved generalization bounds of group invariant/equivariant deep networks via quotient feature spaces." Uncertainty in Artificial Intelligence. PMLR, 2021
> > >
> > > [4] Amir Gholami, Kurt Keutzer, and George Biros. Anode: Unconditionally accurate memory-efficient gradients for neural odes. arXiv preprint arXiv:1902.10298, 2019.
> > >
> > > [5] Jonas Köhler, Leon Klein, and Frank Noé. Equivariant flows: exact likelihood generative learning for symmetric densities. arXiv preprint arXiv:2006.02425, 2020

---

> > > > ### Comment · Reviewer_ZQjr · 2022-11-02
> > > > **response**
> > > >
> > > > I thank the authors for taking the time to reply to my interrogations and remarks, as well as for updating the manuscript.

---

### Review · Reviewer_cJ3b · 2022-10-03

**Summary Of Contributions:**

This paper proposes a new method for modelling invariant probability densities using flow models. The basic idea, following previous work, is to learn an equivariant diffeomorphism. For this purpose new invertible equivariant layers are introduced. In practice, these work only for discrete groups with permutation matrix representations. The paper proves universality of this type of model for modelling invariant densities. Experiments on synthetic data and small images are performed.


**Broader Impact Concerns:**

no concerns

**Requested Changes:**

Please fix all major and minor issues mentioned above

**Strengths And Weaknesses:**

The theoretical part of the paper is quite strong. I particularly like the universality proof for compact groups, which is a fundamental contribution. The method itself is natural enough, though unfortunately it requires working with discrete groups and permutation representations, which is more restrictive than general compact groups. Finally, the experimental section is a bit limited, both in terms of scope (only small image datasets) and results (it seems that more work is required to make it work really well).

Although I think this is a nice paper with a lot of potential I do have some concerns that in my view need to be addressed before publication:

- On the top of page 3 (section 2.1), the explanation of how the group acts on the space of images is not entirely correct. Indeed G might act in this way, but in general it can also act on the feature vectors R^K via some representation of the stabilizer subgroup H. E.g. a 2D vector field would transform as rho(r) f(r^{-1} (x - t)) for some roto-translation g=(r,t) and rho a 2x2 rotation matrix. This is called the representation of SE(2) induced by the representation rho of SO(2).

- The assumption in proposition 2 that R_g is a permutation matrix only works for discrete groups. So it does not work for the main example of SE(2). This is mentioned later in Remark 1. It seems more transparent to change the narrative of the paper to be about discrete subgroups of SE(2), so that this doesn't come as a surprise / disapointment to the reader.

- It is also not clear how the method can deal with input data that does not transform by a permutation representation, since the layers are constrained to have the same input and output representation. For example, in the toy dataset, the data live in R^2, but we consider an action of C16/D16. The standard action of these groups on R^2 by rotations/reflections is not a permutation representation. Are you converting the 2D vectors to a higher dimensional rep? This needs to be explained in the paper.

- Images transform as scalar fields. The flow layers must have the same input/output representations, so the outputs are also scalar fields. Steerable filters for scalar-to-scalar convolutions are isotropic, which is a bit limiting. Is this correct, and how you implement the network for MNIST/CIFAR?

- The details of network architectures are not included.


Some further minor issues:

- It was not clear to me until page 2 that "finite" refers to the depth of the network (finite number of layers), rather than the symmetry group. It would be helpful to say this more explicitly in the abstract.

- The first paragraph of 4.1 is a bit confusing. The operation of vectorizing does not change anything - if the original vector space has a group representation then the vectorized one has an isomorphic representation. Of course one has to remember and use this representation. I don't follow the story about SO(2), SO(4), C4, GL(4). The main message - that one should use the geometric structure (i.e. that the data is a field, acted on by a group via a certain representation) is clear enough though.

- Proposition 1 is only true if h^i is equivariant wrt the same input and output representation of G, called R_g. You use this in equation 9, h^i(R_g x) = R_g h^i(x), whereas general equivariance would be h^i(R_g x) = T_g h^i(x). So prop 1 needs an additional assumption.

- Typo: page 6. "principle benefit" -> "principal benefit"

- In eq 14, the brackets (R_g x)_1:d should be removed as in the equation below, to R_g x_1:d, because R_g is size d x d.

- Sec 4.6: "It is well known in the literature that a linear map that is equivariant muyst be a convolution with steerable kernels". This is only true if the input and output representations are induced representations, i.e. the input/output data transforms as a field over a base space.

- Sec 4.6, it is not correct to say H = Hom(F_n, F_n+1) since H is a subspace. I would use H = Hom_G(F_n, F_n+1).

- Eq 16: not here the two groups G and H used in the definition of induced representation need to be distinguished. For the case G = E(2) we have H = O(2), and the constraint in eq. 16 only pertains to H.

---

> ### Author Response · Authors · 2022-10-12
> **Rebuttal part 1/2**
>
> We thank the reviewer for their thoughtful comments and feedback and we appreciate the fact that the reviewer felt the theoretical part of our paper was “strong” and that the universality proof for compact groups is a “fundamental” contribution. We now answer the main questions and concerns raised by the reviewer.
>
>
> **Q. The explanation of how the group acts on the space of images is not entirely correct.**
> We thank the reviewer for their insightful remark. It is indeed true that in the general case of steerable feature fields the induced group action can be defined as the reviewer suggests (e.g. $f((tr) x) = rho(r) f(r^{-1} (x - t))$). As noted, in the Weiler and Cesa 2021 [1], gray-scale images can be viewed as scalar fields that transform by the rule $f((tr) x) = f(r^{-1} (x - t))$. Even, in the case of RGB images, the group acts independently on each channel, and thus the representation $rho(r)$ is the identity matrix. We understand that this was not sufficiently clear in the main text and we will update to incorporate the more general setting before introducing the setting that is appropriate for natural images that we consider in this work.
>
> **Q. Proposition 2 that $R_g$ is a permutation matrix**
> The reviewer makes an astute observation that due to $R_g$ in proposition 2 is a permutation matrix which means only discrete subgroups of $SE(2)$ can be modeled. However, we would like to politely push back against the assertion that we should change the narrative of the paper as we would like to highlight that our proposed Residual Flows do not have this constraint on representation. As a result, Residual Flows are not limited to just discrete subgroups of $SE(2)$ unlike the $G$-coupling flow. As we outline in remark 1 we believe this restriction on the group when constructing $G$-coupling highlights the innate difficulty of building finite equivariant flows using the coupling architecture. We will further clarify the distinction between our $G$-coupling flow and $G$-residual flow in Remark 1 by explicitly mentioning that $G$-residual flows are not restricted to finite groups.
>
> **Q. Experiments using permutation representation for Discrete subgroups**
> We appreciate the reviewer’s comment about how we can construct permutation representations for $\mathbb{R}^2$ for subgroups such as $C_{16}, D_{16}$ on toy dataset. We first clarify a potential source of misunderstanding that our toy experiments (e.g. spirals, 8gaussians) are only using $G$-residual flows that do not need permutation representations. As a result, we can simply build our layers using any choice of representation—e.g. Regular representation. For CIFAR-10 experiments, the reviewer is indeed correct that there is no permutation representation for rotations on $\mathbb{R}^2$ for groups such as $C_4$. However, as try to explain in section 4.1 embedding our data in a higher dimensional space we can construct a permutation representation in this higher dimensional space and equivariance is valid in this higher dimensional space. In practice, we bypass the restrictive coupling architecture by leveraging regular representation which indeed means that the flow is not equivariant but still has symmetry-based inductive bias. We have updated the experiments section to fully clarify this point as well as added an Appendix with exact architecture details of the $G$-coupling and a $G$-residual flows we employ.
>
> **Q. Input/Output representations transform as scalar fields so steerable filters are isotropic**
> We appreciate the reviewer's concern that because each flow layer must preserve dimensionality we must always use scalar fields. We first recall that our universality results for $G$-residual flows mean we do not in theory have any loss in representational power. In practice, when constructing these flows the residual map is parametrized using the function $h$ which accepts a scalar field but can itself have intermediate representations that are not scalar fields (e.g. we can use regular, irreps, etc…) as long as the output of $h$ also maps to a scalar field. Such a design principle—where intermediate layers can include more complex representations than the trivial one—is common in Steerable neural networks. Finally, we also point out that our equivariance theory is constructed for $N$-dimensional spaces and to apply this to natural images we embed our data into $\mathbb{R}^N$. In practice, this means that we are able to use Steerable-convolutions for subgroups of $SO(2)$ which acts on the base space of $\mathbb{R}^2$ but because of the embedding in $\mathbb{R}^N$ is preserved. We will clarify this point at the beginning of our experiments.
>
> **Q. The details of network architectures are not included.**
> We plan to add an appendix to include the network architectures we use by the end of the rebuttal deadline.

---

> > ### Author Response · Authors · 2022-10-12
> > **Rebuttal Part 2/2**
> >
> > **Minor Issues**
> > Q. Use of the word finite
> > We think Discrete flows are also equally valid but we feared that this would imply that the data domain was discrete. Given, that there is already literature on discrete flows we opted against this potentially confusing name choice. We will clarify the meaning of finite in the abstract.
> >
> > **Q. Comments regarding section 4.1**
> > We use geometric structure provided by natural images as a vehicle for practical implementation of steerable convolution. Our proposed theory in Theorem 2 and universality of $G$-residual flows does require this and works in the embedded higher dimensional space.
> >
> > **Q. Representation type in proposition 1.**
> > Note that the map outputs images to images so both input and output types are the same and thus it is the same group representation $R_g$.
> >
> > **Q. Typos on Page 6 and Eqn 14**
> > We thank the reviewer for catching these typos, which we will fix in the updated manuscript.
> >
> > **Q. Linear map must be convolutions**
> > We agree with this comment, this is a minor technical detail that we will clarify in the updated manuscript.

---

> > > ### Comment · Reviewer_cJ3b · 2022-10-19
> > > **Response**
> > >
> > > Thanks for the clarifications. The issue with isotropic filters / trivial rho is sufficiently clarified; the fact that input/output of h should both be scalar fields is not too bad a restriction as this would also happen in e.g. residual networks. It is much better than needing isotropic filters.
> > >
> > > Re prop 2: I am ok with keeping the narrative as is; this was just a suggestion. However it is important in my view to make it clear early on what will actually be shown and implemented so as to not give the wrong impression.

---

### Comment · Reviewer_deMB · 2022-10-24
**Self-Dox**

Hi, it appears that the names of the authors are given in the updated version of the manuscript.

---

> ### Author Response · Authors · 2022-10-24
> **Revision is now anonymous**
>
> We thank the reviewer for catching this. We have uploaded a revised version immediately which is anonymous again. We apologize for this and hope this is not going to influence the handling of this paper going forward.

---

### Decision · Action_Editors · 2023-01-27

**Recommendation:** Reject

**Comment:**

The reviewers' opinions ranged from strong reject to borderline. On the positive side, two out of three reviewers found the theoretical contribution of interest and appreciated the quality of the exposition. On the negative side, one reviewer questioned the originality of the theoretical contribution, and no reviewer was fully convinced by the empirical results.

In conclusion, it seems that the paper does not fully meet TMLR's criteria for acceptance at this stage. However, TMLR would consider a resubmission that addressed the following concerns:

- The relationship between Theorem 2 and previous work should be clarified. If Theorem 2 has been proven by Katsman et al. (2021), it ought to be attributed to them, with this paper being clear that its own contribution is a new proof. If not, the paper should make clear what the differences are.

- The experimental evaluation should be strengthened, in order to evaluate the proposed models more convincingly and support the experimental findings more conclusively.

- Both the theory and the experiments should clarify whether the proposed flows are equivariant or approximately equivariant. If the former, the experiments should establish that unambiguously. If the latter, the theory should make it clear early on.

**Audience:**

There is a significant amount of interest in building generative models that respect known symmetries, especially in applications of machine learning to science, so the paper is certainly of interest to TMLR's audience. In particular, I believe the following elements are particularly interesting:

- The new proof of the universality of equivariant flows.
- The theorem that residual equivariant flows are universal.
- The combination of steerable CNNs with commonly used flow layers.

I'm less sure about the following:

- I believe the community would be interested in a more thorough experimental evaluation that the one presented here.
- Propositions 1, 2 and 3 are rather straightforward and likely known to those who already study this area (for example, proposition 2 is a small extension of the fully permutation-equivariant coupling layer of Bender et al., 2020). I think the implementation with steerable CNNs is more interesting, so I'm wondering whether the focus should be on that instead.

**Claims And Evidence:**

The paper proposes equivariant implementations of a number of commonly used normalizing-flow layers using the framework of steerable CNNs. On the experimental side, the proposed equivariant flows are assessed for performance and parameter efficiency in modelling symmetric densities. On the theoretical side, equivariant flows (in general) and equivariant residual flows (in particular) are shown to be universal symmetric-density approximators.

On the experimental side, all three reviewers have expressed concerns regarding how convincing the results are. Quoting from the discussion:

Reviewer cJ3b:

"the experimental section is a bit limited, both in terms of scope (only small image datasets) and results (it seems that more work is required to make it work really well)"

Reviewer ZQjr:

"the main weakness of this work is really the empirical demonstration of the practical use of the proposed methods. Section 6 convinced me that the introduced flows can be more parameter efficient than their counterpart but not much more"

"there are limited baselines being assessed [...], CNFs would be particularly relevant since they are being (rightfully) criticised in the introduction"

Reviewer deMB:

"The experiments do not provide a good empirical backing of the central claims"

"The results in Figure 2 are not convincing. For example, the results for the C16 and D16 results for concentric circles do not appear to be invariant with respect to the groups."

The last point is pertinent, as there seems to be a mismatch between the general framing of the paper (flows with guaranteed equivariance) and what the experiments demonstrate (flows with approximate equivariance).

On the theoretical side, Reviewer deMB noted that the statement of Theorem 2 (which the paper presents as one of its central contributions) has already been shown (albeit in a different way) in previous work (Katsman et al. 2021). It therefore seems that the main contribution of the paper is not the statement of the theorem, but rather a new proof of it. This is certainly a notable contribution, but the fact that the theorem already exists in the literature necessitates a revision of the current paper's framing.